# An open label, non-randomized study assessing a prebiotic fiber intervention in a small cohort of Parkinson's disease participants

Deborah A. Hall[1,11], Robin M. Voigt[2,3,4,11], Thaisa M. Cantu-Jungles[3,5], Bruce Hamaker [3,5], Phillip A. Engen [3], Maliha Shaikh[3], Shohreh Raeisi[3], Stefan J. Green[2,3,6], Ankur Naqib[3,4], Christopher B. Forsyth[2,3,4], Tingting Chen [5,7], Richard Manfready[2], Bichun Ouyang[1], Heather E. Rasmussen[3,8], Shahriar Sedghi [9], Christopher G. Goetz[1] & Ali Keshavarzian [2,3,4,10] ✉

A pro-inflammatory intestinal microbiome is characteristic of Parkinson's disease (PD). Prebiotic fibers change the microbiome and this study sought to understand the utility of prebiotic fibers for use in PD patients. The first experiments demonstrate that fermentation of PD patient stool with prebiotic fibers increased the production of beneficial metabolites (short chain fatty acids, SCFA) and changed the microbiota demonstrating the capacity of PD microbiota to respond favorably to prebiotics. Subsequently, an open-label, non-randomized study was conducted in newly diagnosed, non-medicated ($n = 10$) and treated PD participants ($n = 10$) wherein the impact of 10 days of prebiotic intervention was evaluated. Outcomes demonstrate that the prebiotic intervention was well tolerated (primary outcome) and safe (secondary outcome) in PD participants and was associated with beneficial biological changes in the microbiota, SCFA, inflammation, and neurofilament light chain. Exploratory analyses indicate effects on clinically relevant outcomes. This proof-of-concept study offers the scientific rationale for placebo-controlled trials using prebiotic fibers in PD patients. ClinicalTrials.gov Identifier: NCT04512599.

Risk determinants for Parkinson's disease (PD) include both genetic and environmental factors. Whether sporadic or monogenetic in origin, environmental factors may be critical in triggering PD onset in a susceptible host or influencing disease progression. The intestinal microbiota can impact both health and disease, and it is plausible

that the intestinal microbiota (i.e., the microbiota–gut–brain axis) influences PD symptoms, progression, and treatment success[1]. Studies document a disrupted intestinal microbiota community in multiple neurological diseases including PD[2–4]. Divergence of commensal bacteria composition from the microbial communities found

[1]Department of Neurological Sciences, Rush University Medical Center, Chicago, IL, USA. [2]Department of Internal Medicine, Rush University Medical Center, Chicago, IL, USA. [3]Rush Medical College, Rush Center for Integrated Microbiome and Chronobiology Research, Rush University Medical Center, Chicago, IL, USA. [4]Department of Anatomy & Cell Biology, Rush University Medical Center, Chicago, IL, USA. [5]Whistler Center for Carbohydrate Research, Department of Food Science, Purdue University, West Lafayette, IN, USA. [6]Genomics and Microbiome Core Facility, Rush University Medical Center, Chicago, IL, USA. [7]State Key Laboratory of Food Science & Technology, Nanchang University, Nanchang, China. [8]Department of Nutrition and Health Sciences, University of Nebraska, Lincoln, NE, USA. [9]Department of Medicine, Mercer University, Macon, GA, USA. [10]Department of Physiology, Rush University Medical Center, Chicago, IL, USA. [11]These authors contributed equally: Deborah A. Hall, Robin M. Voigt. ✉e-mail: ali_keshavarzian@rush.edu

in healthy individuals (i.e., dysbiosis) is associated with both early and late stages of PD[5,6]. Although there is no specific microbiota signature for PD, patients have a significantly different intestinal microbial composition compared to age-matched controls[7,8]. The PD-associated microbiota are characterized by increased relative abundance of putative pro-inflammatory, Gram-negative lipopolysaccharide (LPS)-producing pathobionts and decreased relative abundance of putative anti-inflammatory, short-chain fatty acid (SCFA)-producing bacteria[9–15]. Taxa-specific changes widely reported include increased relative abundance of family Enterobacteriaceae, genera *Akkermansia*, *Lactobacillus*, *Bifidobacterium*, along with decreased relative abundance of family Lachnospiraceae, and its lower taxonomic hierarchal putative SCFA-producing genus *Faecalibacterium*[5]. Generally speaking, PD patients have a pro-inflammatory dysbiotic microbiota community. These changes in the microbiota can augment systemic and neuroinflammation through several mechanisms including disruption of intestinal barrier integrity. SCFAs are critical in maintaining intestinal barrier integrity insomuch as barrier disruption (i.e., intestinal hyper-permeability) occurs when SCFAs levels are low in the colon[16,17]. Intestinal barrier disruption permits the entry of pro-inflammatory bacterial components like LPS into the systemic circulation and studies demonstrate that LPS can activate microglia and promote neurodegeneration[18,19]. Thus, low levels of SCFA in PD patients may promote intestinal leakiness contributing to neuroinflammation.

Consumption of prebiotic fibers influences microbiota composition and levels of SCFA[20]. Dietary fibers are not hydrolyzed by mammalian enzymes but instead are fermented by bacteria in the gastrointestinal tract. Each bacterial group has a preference regarding physical and chemical characteristics of fibers, and this information can be leveraged to select a mixture of prebiotic fibers that promote the growth of distinct groups of bacteria that produce SCFA[20–23]. The first objectives of this study were to: (1) determine whether prebiotic fibers can increase SCFA production in PD patient microbiota and (2) determine which prebiotics modify the microbiota and increase SCFA using a stool fermentation system. These outcomes were then used to select fibers that were used in a 10-day, open-label, non-randomized study to determine if: (1) daily consumption of the prebiotic mixture for 10 days is tolerable (primary) and (2) safe for use in PD patients (secondary), and (3) impacts PD-relevant biological outcomes including microbiota, SCFA production, LPS-binding protein (LBP), zonulin, stool calprotectin, cytokines, C-reactive protein, high-mobility group box 1, brain-derived neurotrophic factor, and neurofilament light chain. Two groups of PD patients were included in this study: those early in the disease before initiating medication (i.e., newly diagnosed, non-medicated) and those with more advanced disease on levodopa treatment and/or other PD medications (i.e., treated).

## Results

### Fermentation studies

As previously demonstrated in the literature[24,25], PD patients have lower total SCFA than age-matched controls (Fig. 1, Student's *t* test: *P* = 0.004). Based on these data, it was unclear if PD patient stool could produce SCFA to a similar level as controls even when provided with prebiotic fibers. To address this important question, PD patient stool was fermented with different fibers and SCFA production was assessed. The data indicate that PD patient stool can produce SCFA at a level similar to that observed in controls (Fig. 1, Student's *t* test: *P* = 0.387), indicating the potential of prebiotic fibers to increase SCFA in PD patients.

Subsequently, a stool fermentation assessment was performed to determine the ability of different fibers (resistant starch, rice bran, resistant maltodextrin, inulin) to promote the growth of specific bacterial groups and influence SCFA production. Hierarchical

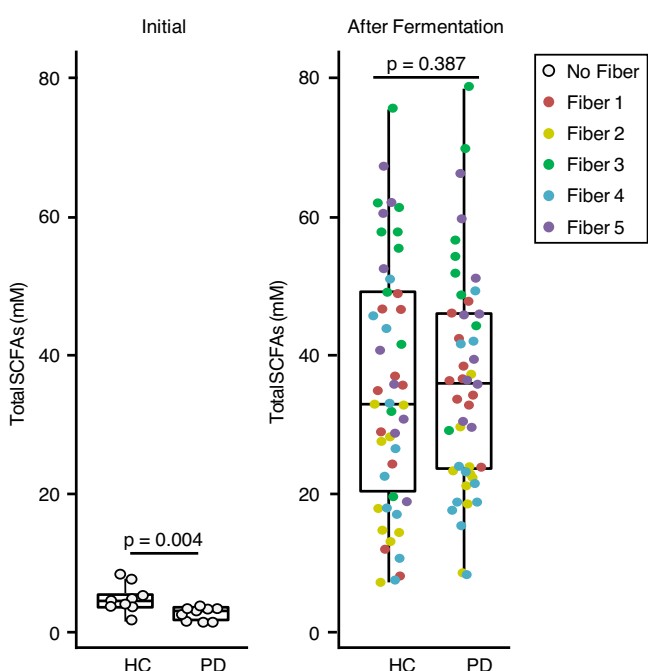

**Fig. 1 | Fermentation of prebiotic fibers with human stool increases short-chain fatty acid (SCFA) production.** Stool obtained from PD patients had a lower concentration of total SCFA than stool obtained from healthy controls (HC) (left panel, Student's *t* test: *P* = 0.004, *n* = 10 biologically independent samples/group). Fermentation of stool with different fibers (Fibers 1–5, 12 h stool fermentation) increased total SCFA production in stool obtained from both HC and PD patients (right panel, Student's *t* test: *P* = 0.387, *n* = 10 biologically independent samples/fiber/group). Fiber 1: fructooligosaccharides (FOS), Fiber 2: glucan, Fiber 3: pectin, Fiber 4: sorghum arabinoxylan and Fiber 5: a mixture of 25% of each of FOS, glucan, pectin, sorghum arabinoxylan. For each box-and-whisker plot, the central horizontal line indicates the median, the bottom and top of edges of the box indicate the 25th and 75th percentiles (respectively), and the top and bottom whiskers indicate the 10th and 90th percentiles (respectively). Each point represents a biologically independent sample. A two-tailed Student's *t* test was used for analysis. Source data are provided as a Source Data File.

clustering and heatmap visualization of stool microbial community structure revealed that each fiber enriched specific bacterial taxa, with limited overlap observed between the fibers. Bacteria enriched by the prebiotic fibers included those from the genus *Prevotella* and families Lachnospiraceae and Ruminococaceae (promoted by resistant starch, Cluster 1); genera *Ruminoccocus*, *Dorea*, and *Bacteroides* (promoted by rice bran, Cluster 2); genera *Blautia*, *Anaerostipes*, and *Bifidobacterium* (promoted by inulin, Cluster 3); and genus *Parabacteroides* (promoted by resistant maltodextrin, Cluster 4) (Fig. 2a).

All fibers promoted SCFA production with fiber-specific effects noted for each SCFA (Fig. 2b–e, one-way ANOVA: total SCFA *P* < 0.0001, acetate *P* < 0.0001, butyrate *P* < 0.0001, Propionate *P* < 0.0001; post hoc analysis was used to identify between-group differences which are summarized in Supplementary Table S1). Resistant maltodextrin and inulin produced the greatest increase in total SCFA levels followed by resistant starch and rice bran (Fig. 2b and Supplementary Table S1). Each fiber type had a distinct metabolic signature: resistant starch was highly butyrogenic, resistant maltodextrin was highly propiogenic, and inulin and rice bran promoted the production of all three SCFA (Fig. 2c–f and Supplementary Table S1). Although rice bran produced less SCFA than the other fibers, it increased the abundance of unique bacteria that were not enriched by other fibers (e.g., *Ruminococcus*, *Dorea*, *Fusobacterium*). Thus, to support the growth of a diverse group of beneficial bacterial groups

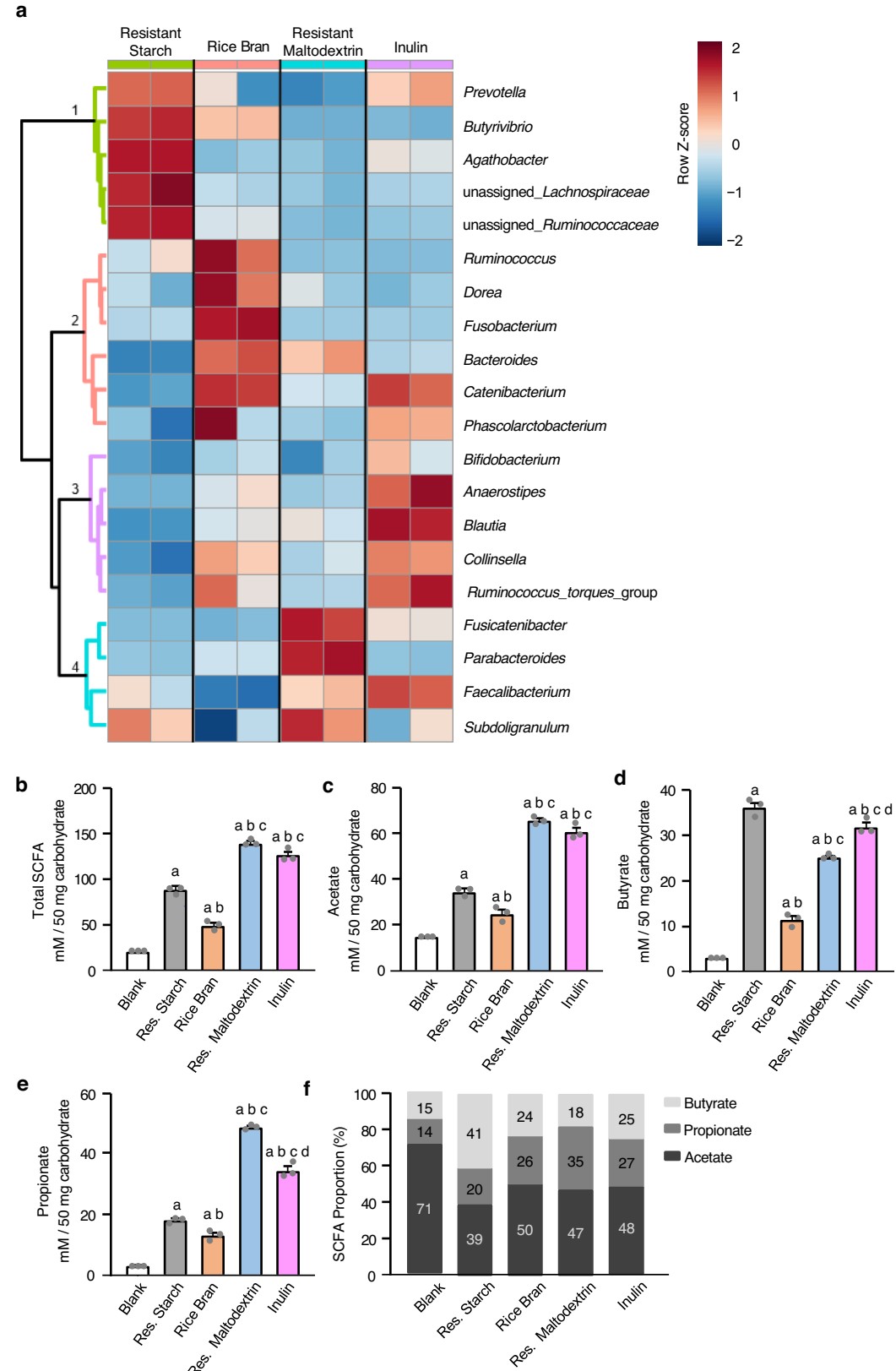

and promote the production of all three SCFA, the following fiber mixture composition was used: 30% resistant starch (raw potato starch), 30% resistant maltodextrin (Nutriose™), 30% stabilized rice bran, and 10% agave branched inulin. These fiber proportions were subsequently incorporated into a bar for testing in participants with PD.

**Tolerability and feasibility of a prebiotic intervention in PD patients**

Participants were moderately accordant to a Mediterranean diet at baseline (median 8.5, IQR 4.3). No differences in accordance existed between newly diagnosed, non-medicated, and treated participants (Mann–Whitney $U$ test: $P = 0.96$). Participants consumed the prebiotic

**Fig. 2 | Fermentation of human stool with prebiotics alters microbial community structure and increases the production of SCFA.** A stool slurry was incubated with fiber (24 h stool fermentation). Three experimental replicates per treatment were conducted, two of which were used for microbiota sequencing (each represented as a separate column). **a** Microbial community structure was assessed using DNA-based 16 S rRNA gene amplicon sequencing ($n = 2$ biologically independent samples). Hierarchical clustering of the 25 most abundant genera are visualized (heatmap represents log2 relative abundance). Hierarchical clustering was performed using Euclidean distances and the Ward algorithm, and clusters of taxa were associated with fiber types. Data are presented as Z-scores of relative abundances normalized within each row. **b**–**e** Short-chain fatty acid-production (mM) during 24 h stool fermentation including total SCFA, acetate, butyrate, and propionate ($n = 3$ biologically independent samples in each group). Error bars represent standard deviation from the mean. One-way ANOVA followed by Tukey's post hoc test was used for analysis, with $P$ values adjusted for multiple comparisons: a = significant vs blank, b = significant vs resistant starch, c = significant vs rice bran, d = significant vs resistant maltodextrin (significance $P < 0.05$, $P$ values for each comparison are reported in Supplementary Table S1). **f** Proportions of each short-chain fatty acid (as a percent of total SCFA) produced during the 24 h stool fermentation ($n = 3$ biologically independent samples). Source data are provided as a Source Data File. Res. Starch resistant starch, Res. Maltodextrin resistant maltodextrin.

bar for 10 days and were subsequently asked how likely they were to continue consuming the prebiotic bar daily with a score ranging from 0 (unlikely) to 10 (highly likely): 55% reported they would be highly likely to continue consuming the bar, 10% likely, 35% somewhat likely, and 0% responded somewhat unlikely or unlikely (Table 1 and Supplementary Table S2) suggesting that consuming the prebiotic bar was both feasible and well-tolerated.

Consumption of prebiotics can influence gastrointestinal symptoms[26,27], thus impact on gastrointestinal symptoms (safety) were assessed at baseline and after the prebiotic intervention (Table 1). Symptoms of bloating and diarrhea were unchanged by the prebiotic intervention in newly diagnosed, non-medicated (Wilcoxon signed-rank test: $P = 0.45$ and $P = 1.0$, respectively) and treated participants (Wilcoxon signed-rank test: $P = 0.50$ and $P = 1.0$, respectively) indicating that the prebiotic fiber mixture did not cause substantial or clinically pertinent gastrointestinal side effects. Analysis of the Gastrointestinal (GI) Symptom and Severity Checklist score revealed that treated PD participants had an improvement in the total GI score after the prebiotic intervention (paired $t$ test: $P = 0.01$), who demonstrated a higher GI symptom score at baseline compared to newly diagnosed, non-medicated PD participants. Regression analyses of the total GI Symptom score (controlling for age, disease duration, levodopa equivalent dose) revealed a difference between baseline and after the prebiotic intervention in the treated PD group (Type III F-test: $P = 0.04$), but no differences were noted for constipation (Type III F-test: $P = 0.23$) or frequency of bowel movements (Type III F-test: $P = 0.54$). Taken together, these results suggest that there were no negative GI side effects associated with the prebiotic intervention, and gastrointestinal symptoms improved in treated PD participants.

As an exploratory analysis, an assessment of UPDRS score was conducted. The 10-day prebiotic intervention was associated with a decrease (Wilcoxon signed-rank test: $P < 0.01$) in total UPDRS score from baseline to after the prebiotic intervention (median 11.50, median 9.0, respectively).

## The prebiotic intervention changed the microbiome

There were no significant differences in the microbiome of non-medicated, newly diagnosed and treated PD participants at baseline (PERMANOVA: $q = 0.28$, PERMDISP: $q = 0.85$, Supplementary Table S3, Supplementary Fig. S1). Alpha-diversity metrics (i.e., Shannon index and Simpson's index) decreased after the prebiotic intervention (Fig. 3a, Wilcoxon signed-rank test: $P = 0.047$ and 0.04, respectively). Although differences in alpha-diversity were noted, total microbial community structure was not significantly impacted by the prebiotic intervention (Fig. 3b, PERMANOVA: $q = 0.43$, PERMDISP: $q = 0.05$, Supplementary Table S3). Nonetheless, the prebiotic intervention was associated with differences in taxon abundance at the species level (Fig. 3c and Supplementary Table S4). The relative abundance of the pro-inflammatory phylum Proteobacteria was decreased by the prebiotic intervention (Wilcoxon signed-rank test: $P = 0.003$, $q = 0.016$; relative abundance (RA): baseline 0.02–16.29%, after intervention 0.00–6.99%) and a decrease was also noted for species *Escherichia coli* (Wilcoxon signed-rank

test: $P = 0.029$, $q = 0.258$; RA: baseline 0.00–15.98%, after intervention 0.00–6.89%) but this difference did not meet the strict criteria for q-value significance (Fig. 3d, e and Supplementary Table S4). Conversely, the relative abundance of putative SCFA-producing species was increased by the prebiotic intervention including *Fusicatenibacter saccharivorans* (Wilcoxon signed-rank test: $P = 0.001$, $q = 0.021$; RA: baseline 0.00–6.60%, after intervention 0.00–16.14%) and *Parabacteroides merdae* (Wilcoxon signed-rank test: $P = 0.003$, $q = 0.044$; RA: baseline 0.00–3.63%, after intervention 0.00–13.08%) (Fig. 3f, g). An increase was noted for the SCFA-producing species *Faecalibacterium prausnitzii* (Wilcoxon signed-rank test: $P = 0.049$, $q = 0.313$; RA: baseline 0.00–19.50%, after intervention 0.00–24.57%), *Bifidobacterium adolescentis* (Wilcoxon signed-rank test: $P = 0.014$, $q = 0.212$; RA: baseline 0.00–40.12%, after intervention 0.00–54.62%), *Ruminococcus bicirculans* (Wilcoxon signed-rank test: $P = 0.007$, $q = 0.212$; RA: baseline 0.00–5.82%, after intervention 0.00–7.22%), but these failed to reach the stringent level of $q$-value significance (Fig. 3h–j and Supplementary Table S4). In addition, it was noted that some SCFA-producing bacteria were reduced by the 10-day prebiotic intervention indicating species-specific differences including *Ruminococcus bromii* (Wilcoxon signed-rank test: $P = 0.014$, $q = 0.212$; RA: baseline 0.00–25.17%, after intervention 0.00–11.83%) *and Ruminococcus torques* (Wilcoxon signed-rank test: $P = 0.021$, $q = 0.218$; RA: baseline 0.00–6.74%, after intervention 0.00–3.06%) but these did not meet the stringent $q$ level of significance (Fig. 3k, l and Supplementary Table S4). Subgroup analysis of newly diagnosed, non-medicated and treated PD participants was conducted separately (Supplementary Table S5).

The prebiotic-induced changes in the microbiota communities were accompanied by significant changes in the abundance of 64 genomic pathways (Wilcoxon signed-rank test: $q < 0.05$, Supplementary Table S6), indicating an overall change in the functional capacity of the microbiota. The prebiotic intervention downregulated multiple biosynthetic pathways previously reported to be upregulated in PD patients (compared to spousal controls)[28]. In particular, the acetyl-CoA fermentation to butanoate II (PWY-5676) is reported to be enriched in PD participants relative to their spouses, but this pathway was downregulated after the prebiotic intervention in PD participants (Supplementary Table S6, Wilcoxon signed-rank test: $q < 0.05$).

Prebiotic intervention-induced changes in the microbiota were associated with a concurrent increase in plasma total SCFA (Wilcoxon signed-rank test: $P = 0.006$, Fig. 3m), as well as each individual SCFA: acetate (Wilcoxon signed-rank test: $P = 0.006$), propionate (Wilcoxon signed-rank test: $P = 0.006$), butyrate (Wilcoxon signed-rank test: $P = 0.059$ and (propionate + butyrate)/total SCFA ratio (Wilcoxon signed-rank test: $P = 0.030$) (Table 2). Subgroup analysis of non-mediated, newly diagnosed, and treated PD participants was assessed separately (Supplementary Table S7).

Overall, the prebiotic intervention reduced the relative abundance of putative pro-inflammatory bacteria and increased the abundance of putative SCFA-producing bacteria in the stool with a concurrent increase in plasma SCFA. The prebiotic intervention also

**Table 1 | Tolerability and gastrointestinal symptoms before and after the prebiotic intervention**

| Tolerability | | | |
|---|---|---|---|
| | Post study survey (n = 20) | | |
| **Would continue the bar, n (%)** | | | |
| Unlikely (0–2) | 0 (0%) | | |
| Somewhat unlikely (3–4) | 0 (0%) | | |
| Somewhat likely (5–6) | 7 (35%) | | |
| Likely (7–8) | 2 (10%) | | |
| Highly likely (9–10) | 11 (55%) | | |
| **Ease of eating 1 bar, n (%)** | | | |
| Difficult (0–2) | 0 (0%) | | |
| Somewhat difficult (3–4) | 0 (0%) | | |
| Neutral (5–6) | 3 (15%) | | |
| Somewhat easy (7–8) | 3 (15%) | | |
| Easy (9–10) | 14 (70%) | | |
| **Ease of eating 2 bars, n (%)** | | | |
| Difficult (0–2) | 1 (5%) | | |
| Somewhat difficult (3–4) | 2 (10%) | | |
| Neutral (5–6) | 2 (10%) | | |
| Somewhat easy (7–8) | 9 (45%) | | |
| Easy (9–10) | 6 (30%) | | |
| **Gastrointestinal symptoms** | | | |
| | Baseline (n = 20) | Prebiotic (n = 20) | P value |
| **All PD (n = 20)** | | | |
| Total score, mean (SD) | 35.85 (25.91) | 27.65 (23.16) | 0.09 |
| Constipation, median (IQR) | 0.5 (5) | 0 (2) | 0.29 |
| Infrequent bowel movements, median (IQR) | 1.5 (5) | 1 (2) | 0.20 |
| **Newly diagnosed, non-medicated PD (n = 10)** | | | |
| Total score, mean (SD) | 29.90 (26.87) | 32.60 (30.18) | 0.63 |
| Constipation, median (IQR) | 0 (2) | 0 (2) | 0.50 |
| Infrequent bowel movements, median (IQR) | 0.5 (2) | 1 (1) | 0.94 |
| **Treated PD (n = 10)** | | | |
| Total score, mean (SD) | 41.8 (24.82) | 22.7 (12.91) | 0.01* |
| Constipation, median (IQR) | 4 (7) | 0.5 (2) | 0.09 |
| Infrequent bowel movements, median (IQR) | 3.5 (7) | 1 (2) | 0.06 |

Tolerability was assessed using a post bar survey and gastrointestinal symptoms were assessed via the PROMIS gastrointestinal symptom scale. The tolerability questionnaire was a Likert scale 1–10. Data are shown as n (%), median (interquartile range, IQR), or mean (standard deviation, SD) as indicated. Numbers used in each analysis are indicated in the. A two-tailed paired t test (mean) or Wilcoxon signed-rank test (median) were used for analysis.
*Significance: P value <0.05. n = 20 PD participants.

influenced the functional capacity of the microbiota and down-regulated a pathway (i.e., acetyl-CoA fermentation) previously reported to be increased in PD patients.

### Prebiotic fiber-induced changes in barrier integrity and inflammation

Plasma zonulin, a putative marker of intestinal barrier integrity, was decreased (Paired t -test: P = 0.001) following the prebiotic intervention (Fig. 4a), an effect that was largely driven by treated PD participants (Paired-t test: P < 0.001, Supplementary Table S7). Systemic levels of zonulin have been suggested as a marker of intestinal barrier integrity and inflammation[24,29] which is congruent with the finding that intestinal inflammation assessed via calprotectin (a marker of neutrophils in the intestinal mucosa) was

reduced after the prebiotic intervention (Paired t -test: P = 0.044, Fig. 4b). Despite changes in the microbiota and markers of intestinal barrier integrity and inflammation there were no prebiotic intervention-induced changes in LBP, serum cytokines (IL-6, IL-8, IL-10, IFN-γ, TNF-α), nor CRP (Table 2).

### Effects of prebiotic fiber mixture on neural and glial-derived proteins

It is well-established that the intestinal milieu can influence the brain, including neuroinflammation, levels of trophic factors, and neurodegeneration. No prebiotic intervention-induced changes were noted for neuroinflammation assessed via HMGB-1 nor for the neurotrophic factor BDNF (Table 2). However, the selected marker of degeneration NfL was reduced after the prebiotic intervention (Paired t test: P = 0.003, Fig. 4c and Table 2). The change in NfL was driven by newly diagnosed, non-medicated PD participants (Paired t test: P = 0.008), although treated PD participants also had reduced NfL following the prebiotic intervention (Supplementary Table S7).

### Correlations of bacterial taxa, biological outcomes, and clinical characteristics

Correlation analyses were conducted to assess the relationship between the relative abundance of taxa, biological outcomes, and demographic and clinical parameters at baseline (Table 3). Zonulin positively correlated with putative pro-inflammatory bacterial species *Escherichia coli* (Spearman's correlation: q = 0.014) (consistent with pro-inflammatory bacteria promoting intestinal barrier dysfunction) and negatively correlated with putative SCFA-producing bacterial species *Parabacteroides merdae* (Spearman's correlation: q = 0.047) (consistent with SCFA-producing bacteria reducing intestinal barrier dysfunction). SCFA-producing bacteria are purported to be anti-inflammatory, and the putative SCFA-producing bacteria *Parabacteroides merdae* negatively correlated with serum TNF-α (i.e., systemic inflammation) (Spearman's correlation: q = 0.045). Age-associated changes in the microbiota are reported in the literature and in this study, age was negatively associated with putative SCFA-producing bacterial species *Eubacterium siraeum* and *Ruminococcus bircirculans* (Spearman's correlation: q = 0.014 for both) (which could contribute to inflammaging). Finally, the levodopa daily dose negatively correlated with the relative abundance of the putative SCFA-producing bacterial species *Bifidobacterium adolescentis* (Spearman's correlation: q = 0.047).

### Discussion

This proof-of-concept study found that 10 days of prebiotic intervention was both well-tolerated and safe in PD patients (primary and secondary outcomes) and decreased total GI symptom severity score in treated PD participants. The prebiotic intervention was also associated with anti-inflammatory shifts in the intestinal microbiota, increased SCFA, reduced calprotectin (intestinal inflammation), reduced zonulin (a putative marker of intestinal barrier dysfunction/inflammation), and a subtle, but statistically significant, reduction in NfL (a marker of neurodegeneration). No changes in LBP, cytokines, CRP, HMGB-1, or BDNF were noted which may reflect the short, 10-day duration of this proof-of-concept study. In addition, exploratory analyses show the prebiotic intervention may be associated with improved clinical outcomes (i.e., gastrointestinal symptoms and UPDRS).

Recent studies highlight the importance of the microbiota-brain axis in PD[30–32] and its scientific anchors include: (1) a pro-inflammatory intestinal microbiota (characterized by low SCFA) can trigger intestinal barrier dysfunction, systemic inflammation, microglial activation, neuroinflammation, and oxidative stress (mechanisms contributing to neurodegeneration and alpha-synuclein misfolding)[33–36]; (2) several diseases and disorders associated with increased risk of PD (e.g., metabolic syndrome, diabetes, inflammatory bowel disease,

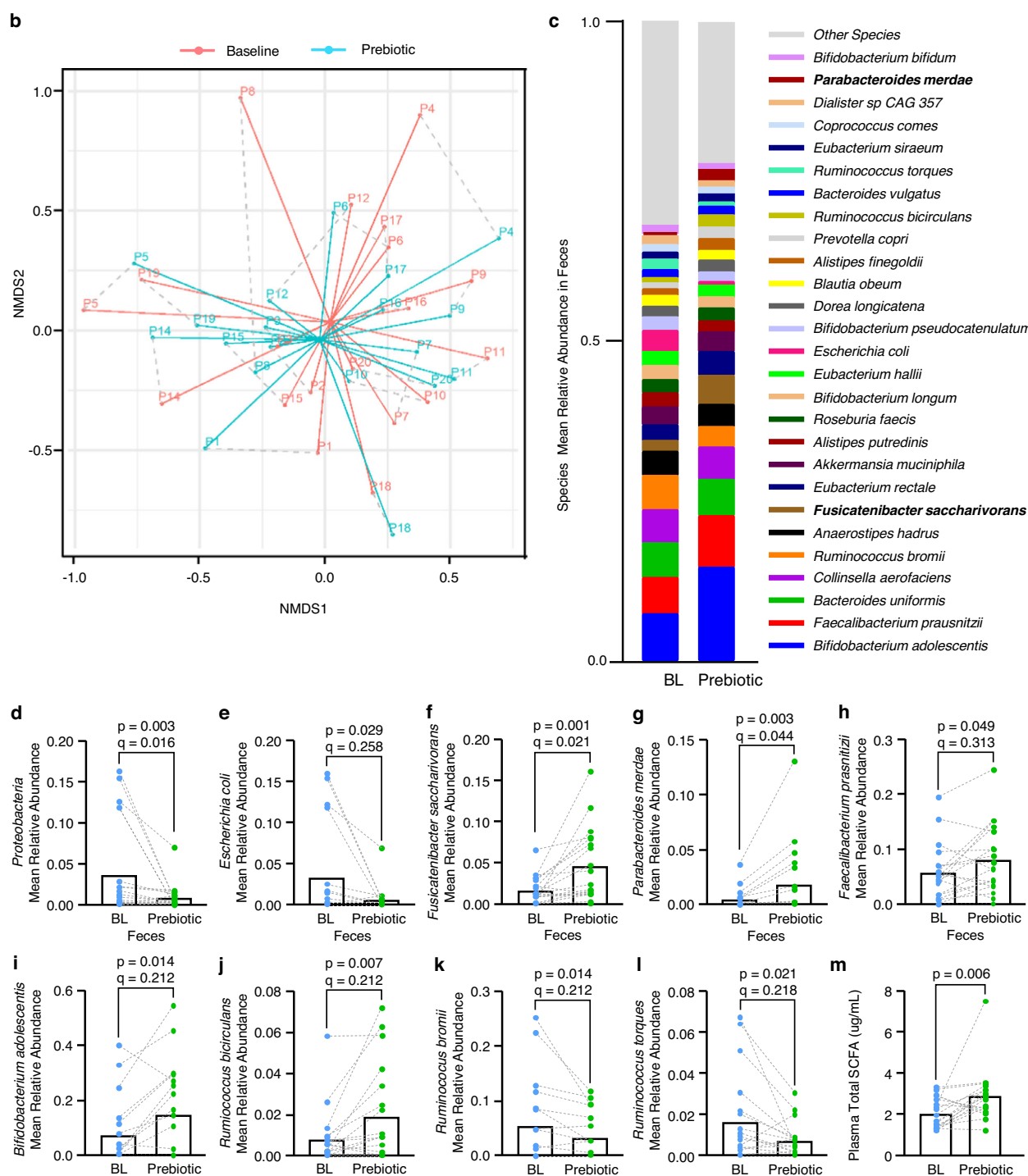

**a**

|  | Baseline Mean ± SD | Prebiotic Mean ± SD | p-Value |
|---|---|---|---|
| Shannon Index | 3.03 ± 0.36 | 2.89 ± 0.31 | 0.047* |
| Simpson's Index | 0.91 ± 0.04 | 0.88 ± 0.06 | 0.040* |
| Species Richness | 74.79 ± 10.11 | 72.26 ± 10.97 | 0.187 |
| Species Evenness | 0.70 ± 0.07 | 0.67 ± 0.07 | 0.120 |

constipation) are associated with microbiota dysbiosis[37–40]; and (3) use of levodopa is associated with reduced relative abundance of putative beneficial, SCFA-producing bacteria. Thus, it is plausible that a prebiotic intervention that alters the microbiota (and increases SCFA) could be an effective strategy to modify disease course, symptoms, or treatment success in PD patients[5].

Studies suggest that microbiota-directed interventions (diet, probiotics, microbiota transplant) may beneficially impact symptoms and/or PD pathogenesis[41]. However, to the best of our knowledge, even though PD is often associated with low SCFA production, no investigations have used a prebiotic mixture designed to augment SCFA production in patients with PD[20]. Prebiotic fibers are generally

**Fig. 3 | Characterization of PD participant stool microbiome at baseline and after the prebiotic intervention. a** Diversity indices, including Shannon Index, Simpson's Index, Species Richness, and Pielou's evenness were calculated at the taxonomic level of species (*n* = 19 biologically independent samples assessed at baseline and after prebiotic intervention). Mean index score and standard deviation (SD) are displayed. **b** Visualization of microbial community structure at baseline and after prebiotic intervention was performed using nonmetric multi-dimensional scaling (NMDS) at the taxonomic level of species (*n* = 19 biologically independent samples assessed at baseline and after the prebiotic intervention). Symbols representing each PD participant (P1–P20) were connected to a centroid representing the mean value of each group: baseline (red) or after prebiotic intervention (blue). A dotted line connects each subject at baseline and after the prebiotic intervention. **c** Mean relative abundance of microbial species (>1% relative abundance) at baseline and after the prebiotic intervention. Bold taxa indicate a significant difference (*q* < 0.05) between baseline and after prebiotic

intervention assessed using a two-tailed, Wilcoxon signed-rank test and corrected for multiple comparisons using the Benjamini–Hochberg method. The prebiotic intervention: **d**, **e** decreased relative abundance of putative pro-inflammatory bacteria from the phylum Proteobacteria and the species *Escherichia coli*; **f**–**j** increased relative abundance of putative beneficial SCFA-producing bacterial species including *Fusicatenibacter saccharivorans*, *Parabacteroides merdae*, *Faecalibacterium prausnitzii*, *Bifidobacterium adolescentis*, *Ruminococcus bicirculans*; **k**–**l** decreased the relative abundance of *Ruminococcus bromii* and *Ruminococcus torques*; and **m** increased levels of plasma SCFA. A two-tailed, Wilcoxon signed-rank test was used for analysis. Bar height represents the group mean and individual samples are indicated. *n* = 19 (**a**–**l**) and *n* = 18 (**m**) biologically independent samples assessed at baseline and after the prebiotic intervention. Group means and standard deviations are shown in Supplementary Table 4 and source data are provided as a Source Data File. BL baseline, Prebiotic after the 10-day prebiotic intervention.

## Table 2 | Biological outcomes before and after the prebiotic intervention

| All PD participants | Baseline | Prebiotic | P value |
|---|---|---|---|
| **Short-chain fatty acids—plasma** | | | |
| Acetate (µg/mL) (*n* = 18) | 1.90 (0.70) | 2.73 (1.29) | 0.006* |
| Propionate (µg/mL) (*n* = 18) | 0.08 (0.05) | 0.12 (0.05) | 0.006* |
| Butyrate (µg/mL) (*n* = 18) | 0.06 (0.04) | 0.08 (0.03) | 0.059 |
| Total SCFA (µg/mL) (*n* = 18) | 2.05 (0.74) | 2.92 (1.31) | 0.006* |
| Butyrate/total SCFA ratio (*n* = 18) | 0.03 (0.02) | 0.03 (0.02) | 0.442 |
| (Propionate + Butyrate)/total SCFA ratio (*n* = 18) | 0.12 (0.06) | 0.15 (0.06) | 0.030* |
| **Intestinal barrier integrity, bacterial translocation, intestinal inflammation** | | | |
| Plasma Zonulin (ng/ml) | 21.08 (6.62) | 13.97 (4.90) | 0.001* |
| Plasma LBP (ng/ml) | 14,476 (5904) | 15,205 (6621) | 0.277 |
| Stool calprotectin (µg/g) | 74.45 (109.00) | 54.95 (86.59) | 0.044* |
| **Systemic inflammation—serum** | | | |
| IFN-γ (pg/ml) | 6.38 (14.02) | 3.36 (2.43) | 0.294 |
| IL-6 (pg/ml) | 0.67 (0.35) | 0.70 (0.47) | 0.840 |
| IL-8 (pg/ml) | 26.72 (75.49) | 10.02 (3.12) | 0.869 |
| IL-10 (pg/ml) | 0.29 (0.22) | 0.33 (0.33) | 0.216 |
| TNF-α (pg/ml) | 0.68 (0.20) | 0.68 (0.23) | 0.856 |
| CRP (mg/L) | 2.20 (4.34) | 2.00 (3.13) | 0.999 |
| **Brain outcomes—serum** | | | |
| BDNF (pg/ml) | 5720 (1422) | 5340 (1701) | 0.277 |
| NfL (pg/ml) (*n* = 19) | 69.62 (34.21) | 63.15 (33.35) | 0.003* |
| HMGB-1 (ng/ml) | 223.46 (106.82) | 209.36 (82.20) | 0.756 |

*BDNF* brain-derived neurotrophic factor, *CRP* C-reactive protein, *HMGB-1* high-mobility group box 1 protein, *IFN-γ* interferon-gamma, *IL* interleukin, *LBP* lipopolysaccharide-binding protein, *NfL* neurofilament light chain, *TNF-α* tumor necrosis factor-alpha.
*Significance: *P* value <0.05. *n* = 20 PD participants, unless noted above.
All data are shown as mean (standard deviation).
Based on the Shapiro-Wilks normality test, either a two-tailed, parametric paired *t* test or a two-tailed, non-parametric Wilcoxon signed-rank test was used for analysis.

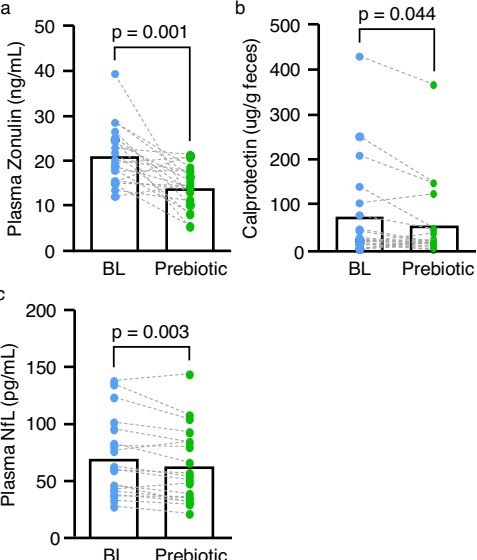

**Fig. 4 | Ten days of the prebiotic intervention had effects on biological outcomes.** Ten days of the prebiotic intervention: **a** reduced plasma zonulin *n* = 20, **b** reduced stool calprotectin *n* = 20, and **c** reduced plasma neurofilament (NfL) *n* = 19. Bar height represents the group mean and individual points are indicated for each group, biological independent samples assessed at baseline and after the prebiotic intervention. A two-tailed, Paired *t*- test was used for analysis. Group means and standard deviations are shown in Table 3, and source data are provided as a Source Data File.

regarded as safe, have been used for centuries to treat chronic illnesses, and are capable of modifying microbiota communities[20,42]. The prebiotic fibers used in this study (resistant starch, rice bran, resistant maltodextrin, inulin) were selected based on physical and chemical properties that make them slow fermenting which is ideal for use in humans. A slow fermentation profile is optimal for a prebiotic, because it is associated with slow generation of gas which is potentially linked with higher tolerability[17,33–38] and indeed, participants in the current study reported no negative GI side effects. In addition, PD patients demonstrate intestinal barrier disruption throughout the gastrointestinal tract (including in the colon)[43–45] and slow fermentation allows more fiber to reach the distal colon as opposed to being

consumed by bacteria in the proximal gastrointestinal tract[17,33–38]. Therefore, barrier disruption in the colon in PD patients can best be addressed with a slow fermenting fiber and could have important PD-relevant effects including reduced inflammation.

As hypothesized based on the fermentation experiment, consumption of the prebiotic mixture altered the stool microbiota community in PD participants including reduced relative abundance of putative pro-inflammatory bacteria (e.g., Proteobacteria) and increased relative abundance of putative anti-inflammatory bacteria including SCFA-producing bacteria (e.g., *Fusicatenibacter saccharivorans*, *Parabacteroides merdae*). These changes were accompanied by an increase in plasma SCFA. Our group and others have demonstrated that PD patients have dysbiotic microbiota characterized by low abundance of SCFA-producing bacteria and intestinal barrier dysfunction (urinary sugar test, immunohistochemical analysis of tight junction proteins in intestinal tissue), LBP, LPS, and zonulin)[29,45–49]. This study used plasma zonulin as a putative marker of intestinal barrier integrity and found that 10 days of the prebiotic

**Table 3 | Significant associations between study outcomes**

| Variable 1 (*n* = 20) | Variable 2 (*n* = 19) | *R*-value | *P* value | *q* value |
|---|---|---|---|---|
| Zonulin | *Escherichia coli* | 0.790 | 0.005* | 0.014* |
| | *Parabacteroides merdae* | −0.660 | 0.043* | 0.047* |
| Tumor necrosis factor-alpha (TNF-α) | *Parabacteroides merdae* | −0.697 | 0.030* | 0.045* |
| Age of PD Participants | *Eubacterium siraeum* | −0.874 | 0.007* | 0.014* |
| | *Ruminococcus bircirculans* | −0.768 | 0.003* | 0.014* |
| Levodopa daily dosage (LEDD) | *Bifidobacterium adolescentis* | −0.618 | 0.047* | 0.047* |

Data were transformed using Log2-fold change from baseline. Spearman's rank correlation coefficient (*R*-value) was used to assess the strength and direction of the association, ranging from values +1 to −1. *P* values were corrected for multiple comparisons using the Benjamini–Hochberg method.
*Significance: *q* value <0.05. *n* = 20 for Variable 1, *n* = 19 for Variable 2.

intervention decreased plasma zonulin in PD patients suggesting that prebiotic treatment may improve intestinal barrier integrity. However, there is controversy regarding the use of serum zonulin as a marker of barrier integrity[50,51] and studies using additional assessments of the intestinal barrier are needed to confirm the conclusion that the prebiotic intervention improves barrier integrity in PD patients. Such studies are important because the barrier is one of many mechanisms by which the prebiotic may impact PD-relevant outcomes.

Microbiota dysbiosis and intestinal barrier dysfunction are often associated with intestinal inflammation. In this study, stool calprotectin (a reliable and specific marker of inflammation) was used to assess intestinal inflammation. Prior studies demonstrate that calprotectin is increased in PD patients[24,29,46,52], and the current study found that stool calprotectin in PD participants was decreased by the prebiotic intervention. While intestinal inflammation (i.e., calprotectin) was reduced by the 10-day prebiotic intervention, it was not sufficient to alter systemic inflammation including LBP, CRP, or cytokines.

The prebiotic intervention modestly but significantly reduced NfL, a putative systemic marker of neurodegeneration[53]. The data in this report suggest that the 10-day prebiotic intervention likely impacts NfL through a mechanism that is independent of systemic inflammation. Based on the results of the fermentation study and the observed increase in serum SCFA, we posit that the effects of the prebiotic intervention were mediated via SCFA. However, multiple mechanisms may be mediating the effect of the prebiotic intervention and will require additional investigation.

Lastly, differences were observed between newly diagnosed, non-medicated and treated PD participants. Because PD is a highly heterogeneous disease, future studies are needed to begin to understand which populations can benefit most from a prebiotic intervention. For example, levodopa-treated PD patients may particularly benefit from a prebiotic intervention. The PD-associated microbiota is characterized by a high abundance of bacteria that contain the levodopa metabolizing enzyme, tyrosine decarboxylase (TDC)[54–56] and the prebiotic intervention may influence the relative abundance of TDC-containing bacteria and positively impact treatment efficacy and reduce dyskinesias in levodopa-treated PD patients. Additional studies with a larger sample size and a randomized, double-blind, placebo-controlled study design are required to confirm the intriguing results of this proof-of-concept study demonstrating that prebiotic intervention was associated with improvement in clinical symptoms (UPDRS). Admittedly, the study was open-label and is subject to participant rater bias but is nonetheless encouraging for further study in more controlled conditions.

A strength of this study is the within-subject design, but there are also limitations including small sample size and an open-label, non-placebo design. The prebiotic mixture was administered in the form of a healthy bar containing additional non-prebiotic fiber with potential antioxidant effects. Thus, the observed biological changes might not be due to prebiotic fibers alone but could instead be related to the consumption of another healthy component in the bar. In addition, while participants were instructed to maintain their usual diet throughout the study, consumption of the prebiotic bar may have influenced diet. The non-blinded design of the study could have impacted the results of clinical outcomes like the UPDRS and gastrointestinal symptom severity score reported by participants (i.e., placebo effect). However, the non-blinded design should not have impacted biological-based outcomes which were analyzed by staff blinded to time (baseline, after prebiotic intervention) and PD group (newly diagnosed, non-medicated vs treated).

This proof-of-concept study demonstrates that a SCFA-promoting prebiotic fiber mixture can be used to modulate the intestinal microbiota in PD patients (i.e., the approach is feasible) and that the prebiotic mixture is well-accepted, tolerated, and safe for use in PD patients. Moreover, the prebiotic fiber mixture may have a clinical impact leading to reduced severity of motor and non-motor PD symptoms and improved gastrointestinal function. This proof-of-concept study provides the scientific rationale for future studies evaluating the potential of a microbiota-directed prebiotic intervention as a disease-modifying therapeutic approach in PD patients.

## Methods

All research activities were approved by the Rush University Medical Center (RUMC) Institutional Review Board (IRB). Consent was obtained from all participants for sharing of de-identified data.

### Dietary fiber impacts microbiota structure and function

**Participants.** Stool samples were obtained from an IRB-approved repository including human stool from healthy control (*n* = 10) and PD (*n* = 10) participants (not the same participants as those enrolled in the clinical study (described below)). There were no significant differences in demographic characteristics between the healthy control and PD groups (age: two-tailed *t* test: *P* = 0.368; sex: chi square analysis: *P* = 0.350, race: chi square analysis: *P* = 0.211). At the time of stool collection, no participants were taking levodopa (LD) or other PD medications, participants did not consume probiotics, synbiotics, or antibiotics within three months prior to sample collection, and no participants were consuming a special diet (e.g., vegan, vegetarian, Paleo). A movement disorder specialist examined and confirmed PD diagnosis. All participants donating samples to the repository signed the RUMC IRB-approved informed consent form (ORA#: 07100403). Participants were not compensated for donating samples to the repository.

**Procedure.** Prebiotic fibers (inulin, resistant starch, resistant maltodextrin, rice bran) were incubated with human stool from healthy control (*n* = 10) or non-medicated PD (*n* = 10) participants. A 12 or 24 h human stool fermentation with each individual fiber was performed[57,58]. A carbonate-phosphate buffer was prepared and sterilized (autoclaved at 121 °C, 20 min). The buffer was then cooled to room temperature, oxygen was removed by bubbling with carbon dioxide, and cysteine hydrochloride (0.25 g/L of buffer) was added as a reducing agent. The buffer was then placed into an anaerobic chamber

the day before experimentation to complete buffer reduction. On the day of the experiment, freshly collected stool samples from three healthy human donors (10 g each) were homogenized with carbonate-phosphate buffer (1:3 [wt/vol]), followed by filtration through four layers of cheesecloth. Then, 1 ml of the pooled stool inoculum was added to Balch-type tubes containing 50 mg of dietary fiber substrate (or without fiber, i.e., blank/negative control) and 4 ml of carbonate-phosphate buffer. Tubes were closed with butyl rubber stoppers, secured with aluminum seals, and incubated for 24 h (on a shaker inside an incubator (37 °C)). All sample manipulation was conducted in an anaerobic atmosphere (85% $N_2$, 5% $CO_2$, and 10% $H_2$). Stool fermentation experiments were conducted in triplicate.

**Microbial community characterization of fermented stool.** Aliquots (1 ml) of fermented stool samples were centrifuged (20,784 g, 15 min), and the supernatant discarded. Automated DNA extraction of the pelleted material was performed using a QIAcube Connect instrument with the QIAamp PowerFecal Pro DNA kit (Qiagen, Germantown, MD) per manufacturer instructions. Recovered genomic DNA was used as a template for amplification of the V4 variable region of microbial 16 S ribosomal RNA (rRNA) genes using primers 515 F (GTGCCAGCMGCCGCGGTAA) and 806 R (GGACTACHVGGGTWTCTAAT) (custom primers from Integrated DNA Technologies, Coralville, IA). Amplicons were generated using a two-stage PCR amplification protocol[59,60]. Briefly, the 515 F and 806 R primers contained 5′ common sequence tags (known as common sequences 1 and 2 [CS1 and CS2]). First-stage PCR amplifications were performed in 10 µl reaction mixtures in 96 well plates, using MyTaq HS 2× master mix (Bioline, Memphis, TN). PCR conditions were 95 °C for 5 min, followed by 28 cycles of 95 °C for 30 s, 55 °C for 45 s, and 72 °C for 60 s. Subsequently, a second PCR amplification was performed in 10 µl reaction mixtures in 96 well plates. Each well received a separate primer pair with a unique 10 base barcode, obtained from the Access Array Barcode Library for Illumina (Fluidigm, South San Francisco, CA; catalog no. 100-4876). These Access Array primers contained the CS1 and CS2 linkers at the 3′ ends of the oligonucleotides. Cycling conditions were 95 °C for 5 min, followed by eight cycles of 95 °C for 30 s, 60 °C for 30 s, and 72 °C for 30 s. Samples were then pooled in equal volumes using an EpMotion5075 liquid handling robot (Eppendorf, Hamburg, Germany). The pooled library was purified using an AMPure XP cleanup protocol (0.6×, vol/vol; Agencourt, Beckman-Coulter, Indianapolis, IN) to remove fragments smaller than 300 bp. Libraries, with a 20% phiX spike-in, were loaded onto two MiniSeq flow cells and sequenced (2 × 153 paired-end reads). Fluidigm sequencing primers, targeting the CS1 and CS2 linker regions, were used to initiate sequencing. Demultiplexing of reads was performed on the instrument. Library preparation and 16 S rRNA gene sequencing were performed at the DNA Services Facility at the University of Illinois at Chicago (Chicago, IL). For each fermentation condition, DNA was extracted from two replicate samples. Raw sequence data (FASTQ files) were deposited in the National Center for Biotechnology Information (NCBI) Sequence Read Archive (SRA), under the BioProject identifier PRJNA852512.

**Fermented stool SCFA analysis.** Triplicates of fermented stool samples were prepared and were analyzed at Purdue University using a gas chromatograph (GC-FID 7890 A; Agilent Technologies Inc.) on a fused silica capillary column (Nukon Supelco no. 40369-03 A; Bellefonte, PA)[58] under the following conditions: injector temperature of 230 °C, initial oven temperature at 100 °C, and temperature increase of 8 °C/min to 200 °C with a hold for 3 min at final temperature. Helium was used as a carrier gas at 0.75 ml/min. Quantification was performed based on relative peak area using external standards of acetate (A38S),

**Table 4 | Parkinson's disease participant demographics**

| Demographic and clinical variables | Newly diagnosed, non-medicated PD (n = 10) | Treated PD (n = 10) | P value |
|---|---|---|---|
| Age, mean (SD) | 62.90 (6.89) | 65.70 (9.03) | 0.45 |
| **Sex, n (%)** | | | |
| Men | 5 (50%) | 6 (60%) | 1.00 |
| Women | 5 (50%) | 4 (40%) | |
| **Race & ethnicity** | | | |
| White, n (%) | 10 (100%) | 10 (100%) | 1.00 |
| Not Hispanic or Latino, n (%) | 10 (100%) | 10 (100%) | 1.00 |
| **Disease characteristics** | | | |
| Age of disease onset, mean (SD) | 61.30 (6.01) | 59.50 (9.96) | 0.63 |
| Disease duration, mean (SD) | 1.95 (1.30) | 6.20 (4.51) | 0.01* |
| H&Y, mean (SD) | 2 (0) | 2 (0) | 1.00 |
| UPDRS motor, mean (SD) | 12 (5.09) | 14.9 (5.62) | 0.24 |
| **Levodopa equivalent dose** | | | |
| Median (IQR) | NA | 393.75 (333.00) | – |
| Range | NA | 100–1387.5 | – |
| Bristol Stool Score, mean (SD) | 3.20 (1.54) | 2.20 (1.75) | 0.19 |

*IQR* interquartile range, *H&Y* Hoehn and Yahr staging scale, *NA* not applicable, *SD* standard deviation, *UPDRS* Unified Parkinson's Disease Rating Scale.
*Significance: P value <0.05. n = 20 PD participants.
Two-tailed Student's t test or chi-square test were used for analyses.

propionate (A258), and butyrate (AC108111000) and an internal standard of 4-methylvaleric acid (AAA1540506) from Fisher Scientific (Hampton, NH).

**Clinical study: study design, participants, and data/sample collection**

**Participants.** All participants signed the RUMC IRB-approved informed consent form (ORA#: 20072703), and the study was registered (ClinicalTrials.gov Identifier: NCT04512599). Participants were not compensated for participating in the study. The study was an open-label, non-randomized study in PD participants (n = 20) conducted at RUMC and included newly diagnosed, non-medicated PD participants (n = 10) and treated PD participants (n = 10) receiving levodopa (LD) and/or other PD medications. A movement disorder specialist (Drs. Hall or Goetz) examined and confirmed the diagnosis of PD patients. Parkinsonian symptoms were assessed using the Unified Parkinson's Disease Rating Scale (UPDRS) Part 3[61], and Hoehn and Yahr (H&Y) staging scale[62]. Participant characteristics are shown in Table 4 including: age of onset, disease duration, motor UPDRS, H&Y, levodopa equivalent daily dosages (LEDD), Bristol stool score, and demographic data (i.e., age, sex, race, ethnicity). Inclusion criteria were: current diagnosis of PD (UK Brain Bank Criteria, H&Y stages 1–4 inclusive)[63], age (>30), and able to consent. Exclusion criteria included: (1) intestinal resection, (2) history of gastrointestinal disease except for hiatal hernia, GERD, or hemorrhoids, (3) severe renal disease defined by creatinine more than 2.5 times normal, (4) abnormal liver function defined by ALT/AST > 4 times normal or elevated bilirubin, (5) antibiotic use within the 12 weeks prior to enrollment, (6) consumption of probiotics, prebiotics, or synbiotics within four weeks prior to enrollment, (7) consumption of a non-standard diet (e.g., vegan, vegetarian, gluten-free, Paleo).

**Design.** Each participant had a baseline visit and a follow-up visit after 10 days of the prebiotic intervention. Participants consumed the

prebiotics in the form of a bar containing resistant starch, rice brain, resistant maltodextrin, and inulin for 10 days (one bar = 10 g fiber). One bar was consumed daily during the first three days, and then one bar twice a day for an additional seven days. Participants were instructed to consume the prebiotic bar in the morning for the first three days, then in the morning and afternoon for the following week (they were not instructed to take it separately from meals). The prebiotic mixture was a proprietary formula developed by Drs. Keshavarzian, Hamaker, and Cantu-Jungles. The prebiotic bar was created by Drs. Keshavarzian, Hamaker, Cantu-Jungles, Sedghi, and Rasmussen and was produced by a licensed packing company: Gramercy Bakery, LLC. Ingredients of the bar were organic, generally recognized as safe, food-grade ingredients.

**Data collection.** The UPDRS and H&Y were used to assess PD characteristics. Diet quality was assessed at baseline using the validated, 14-item Mediterranean Diet Adherence Screener (MEDAS)[64] and participants were asked to continue their usual diet during the 10-day study. A questionnaire was created to gauge the tolerability and feasibility of consuming the bar. Questions included: satisfaction with taste, portion, and texture, whether it decreased appetite, if they liked the bar, wanted more, or would continue it for 6–12 months, and ease of taking 1–3 bars per day which were scored on a scale from 0 to 10. Participants completed the NIH Patient-Reported Outcomes Measurement Information Systems (PROMIS) questionnaire[65] to assess the impact of the prebiotic intervention on gastrointestinal symptoms including bowel movements, stool consistency, abdominal discomfort/pain, bloating, and flatulence on a scale from 1 (best) to 10 (worst).

**Sample collection.** Participants collected stool at home 12–24 h before the baseline and end of study visits using an anaerobic collection kit (BD Gaspak, Becton Dickinson and Company, Sparks, MD)[66]. Blood was collected at baseline and end of the study and was processed for serum and plasma within one hour of collection and was stored at −80 °C until analysis.

**Microbial community characterization.** Stool microbiome was assessed using non-targeted shotgun metagenome sequencing and taxonomic and functional gene profiling. Total DNA was extracted from stool samples (250 mg) utilizing the FastDNA bead-beating Spin Kit for Soil (MP Biomedicals, Solon, OH, USA), and verified with fluorometric quantitation (Qubit 3.0, Life Technologies, Grand Island, NY). Library preparation was performed using a Swift 2 Turbo DNA Library kit (Swift Biosciences, Ann Arbor, MI) with 50 ng of input DNA and five cycles of PCR for indexing with unique dual indices. Libraries were sequenced on an Illumina NovaSeq6000 instrument employing an SP flow cell (paired-end 2 × 150 base reads). Libraries were created in the Genomics and Microbiome Core Facility at Rush University Medical Center, and sequencing was performed at the DNA Services Lab at the University of Illinois at Urbana-Champaign. Raw FASTQ files were checked for quality using FastQC (v 0.11.9). Sequence reads were quality filtered and trimmed using the algorithm bbduk (Department of Energy Joint Genome Institute)[67]. Filtered and trimmed data were again checked for quality using FastQC (v 0.11.9). Taxonomic profiling was generated with MetaPhlAn3 (v3.0.7) and functional profiling was performed using the software package HUMAnN3 (v3.0.0.α.3) mapping to the UniRef90 catalog (UniRef release 2019_01)[68]. Taxonomic and functional gene pathway biological observation matrices (BIOMs) are provided in the Source Data File. UniRef90 relative abundance tables were regrouped into the following higher-level organizations: MetaCyc pathways, KEGG orthology, and UniProt gene families. All the downstream data processing (i.e., alpha-diversity, beta-diversity, functional pathways, PERMANOVA, Spearman's correlations, and centroid-based NMDS plots) were performed using open-source packages within the R programming language. Raw sequence data

(FASTQ files) were deposited in the National Center for Biotechnology Information (NCBI) Sequence Read Archive (SRA), under the BioProject identifier PRJNA756556.

**SCFA analysis.** SCFA analyses were conducted at the Proteomics and Metabolomics Facility of Colorado State University (Fort Collins, CO) using gas chromatography-mass spectrometry (GC-MS)[69]. Plasma (200 µL) was added to 30 µL of cold internal standard solution containing 50 µg/mL of $^{13}C_2$-acetate in 2 N HCl. Samples were vortexed (30 s), followed by the addition of 0.35 mL of cold methyl tertiary-butyl ether (MTBE), and were again vortexed (15 s). Samples were centrifuged (3000 × g, 10 min, 4 °C) and the top MTBE layer was recovered and stored at −20 °C until analysis. The MTBE extract (1 µL) was injected into a Trace 1310 GC coupled to a Thermo ISQ-LT MS, at a 5:1 split ratio from Thermo Fisher Scientific (Waltham, MA). The inlet was held at 240 °C. SCFA separation was achieved on a 30 m DB-WAXUI column (J&W, 0.25 mm ID, 0.25-µm film thickness). Oven temperature was held at 100 °C for 0.5 min, ramped at 10 °C/min to 175 °C, then ramped to 240 °C at 40 °C/min, and held at 240 °C for 3 min. Helium carrier gas flow was held at 1.2 mL/min. Temperatures of transfer line and ion source were both held at 250 °C. SIM mode was used to scan ions 45, 60, 62, 73, 74, 88 at a rate of 10 scans/s under electron impact mode. Data processing was completed using Chromeleon software (v 7.3.1, Thermo Fisher Scientific). Retention times were used to differentiate acetate, butyrate, and propionate from other metabolites (e.g., branch-chain fatty acids, isobutyric acid, isovaleric acid, isopropionate)[69]. The limit of detection (LOD) range was 0.3–0.6 µg/ml for acetate and 0.03–0.12 µg/ml for propionate and butyrate.

**Stool calprotectin.** Stool calprotectin is a reliable and widely used marker to assess intestinal inflammation, including in PD[29,46,52]. There is a direct correlation between calprotectin levels and intestinal inflammation assessed via both endoscopy and histology[70] and calprotectin is the gold standard used to monitor intestinal inflammation and assess response to treatment in patients with inflammatory bowel disease (IBD)[71]. ELISA was used to examine stool calprotectin (BÜHLMANN fCAL® ELISA-EK-CAL; BUHLMANN Diagnostics Corp, Amherst, NH, range: 30–1800 µg/g, samples were run in duplicate according to the manufacturer instructions, and data were all within the range of the assay).

**Zonulin.** Zonulin modulates the intestinal barrier and is proposed as a marker of intestinal barrier integrity, is linked to intestinal inflammation, and is reported to be increased in PD patients[24,29,46,50,72,73]. Additionally, zonulin levels are increased in patients with chronic inflammatory diseases known to have disrupted intestinal barrier integrity (i.e., COVID-19, diabetes, irritable bowel syndrome (IBS))[73–76]. A commercially available ELISA kit was used to assess plasma zonulin (MBS706368, MyBioSource, range: 0.625–40 ng/ml, all samples were run in duplicate and data were within the range of the assay).

**Lipopolysaccharide-binding protein.** Lipopolysaccharide (LPS) binding protein (LBP) is widely used as a marker of bacterial translocation and inflammatory response to exposure to LPS (i.e., LPS binds to LBP to form a complex that subsequently binds to CD14 to elicit an immune response)[77–80]. LBP levels in serum are increased during sepsis and LBP levels are upregulated in diseases associated with chronic, low-grade inflammation induced by microbiota dysbiosis and intestinal barrier dysfunction including in PD patients[47–49,81]. LBP was evaluated in this study using ELISA (HK3151, Hycult Biotech, range: 4.4–50 ng/ml, samples diluted 1:1500, all samples were run in duplicate, and data were within the range of the assay).

**Cytokines.** Serum inflammatory cytokines were assessed using the V-PLEX Pro-inflammatory Panel 1 Human Kit including interferon-

gamma (IFN-γ), interleukin (IL)−6, IL-8, IL-10, and tumor necrosis factor-alpha (TNF-α) (K15049D, Meso Scale Diagnostics, LLC, Rockville, MD, USA, range: IFN-γ: 0.37−938 pg/ml, IL-6: 0.06−488 pg/ml, IL-8: 0.07−375 pg/ml, IL-10: 0.04−233 pg/ml, TNF- α: 0.04−248 pg/ml, all samples were run in duplicate and data were within the range of the assay).

**C-reactive protein.** C-reactive protein (CRP) is an acute phase protein that is reliably used to assess inflammation and was assessed by Labcorp (Labcorp, Dublin OH, (range 0–10 mg/L).

**High-mobility group box 1 protein.** Serum high-mobility group box 1 protein (HMGB-1) was examined as a neuroinflammatory biomarker (ELISA NBP2-62766, NOVUS Biological, range: 0.031−2 ng/mL, samples diluted 1:500, all samples were run in duplicate and data were within the range of the assay)[82].

**Brain-derived neurotrophic factor.** Brain-derived neurotrophic factor (BDNF) is a neurotrophic factor that supports the survival and function of neurons and serum BDNF was assessed (K1516WK, U-PLEX, Meso Scale Diagnostics, LLC, range: 0.72−2000 pg/mL, samples diluted 1:4, samples were run in duplicate, four samples were above the highest detection limit of the assay (2000 pg/mL) and were set to the assay maximum of 2000 pg/mL for analysis)[83,84].

**Neurofilament light chain.** Serum neurofilament light chain (NfL) was evaluated as a biomarker of neurodegeneration (F217X, R-PLEX, Meso Scale Diagnostics, LLC, range: 5.5−50,000 pg/ml, all samples were run in duplicate and data were within the range of the assay)[85,86].

### Statistical analysis

Demographics, including age, sex, PD treatment status, UPDRS, H&Y stage, and disease duration were summarized using descriptive statistics. The outcomes for this proof-of-concept study were tolerability and feasibility, safety and gastrointestinal symptoms, and biological outcomes. This study was not powered to assess PD motor symptoms. Changes in gastrointestinal symptoms and side effects (e.g., bloating, diarrhea) from baseline to after the prebiotic intervention were assessed with a two-tailed, paired $t$ test or Wilcoxon signed-rank test. Regression analysis was performed for changes in gastrointestinal symptoms with group, age, disease duration, and levodopa equivalent dose were included in the model. Exploratory analyses were used to compare the UPDRS motor scores between baseline and after prebiotic intervention using a two-tailed, paired $t$ test. Biological outcomes (i.e., SCFA, intestinal barrier integrity, inflammation, NfL, etc) were analyzed using either a two-tailed, paired $t$-test or a two-tailed, Wilcoxon signed rank test to assess changes between baseline and after the prebiotic intervention. Analyses were conducted using the software GraphPad Prism (v9.0, GraphPad Software LLC, San Diego, CA). A $P$ value <0.05 was considered significant. The Shapiro−Wilks test of normality was conducted prior to performing analyses. Adjustments for multiple comparisons ($q$ values) were applied as indicated.

Microbial profiling of fermented stool samples using 16 S rRNA gene amplicon sequencing and SCFA data analysis was performed as follows. Demultiplexed and preprocessed 16 S rRNA gene sequence reads were supplied as raw paired-end FASTQ sequence files. The raw files were imported and processed using the software package QIIME2[87] (q2) version 2020-2. Amplicon sequence variants (ASVs) were generated using DADA2 with sequences trimmed at 153 base pairs. Sequence alignment and construction of a phylogeny tree were obtained using the QIIME2 pipeline align-to-tree-mafft-fasttree and taxonomic assignment was carried out using the q2-feature-classifier plugin against the SILVA SSU rRNA reference database[88] with 99% similarity specific for the V4 16 S region. The number of reads from each sample was rarefied to 5563 sequences/sample for downstream analysis. Relative abundances

were obtained by collapsing rarefied ASVs at the taxonomic level of genus for the different substrate ferments. Differences among microbial groups were visualized through hierarchical clustering and heatmap analysis, using Ward's algorithm and merging clusters based on Euclidean distances through the MicrobiomeAnalyst web-based tool (https://www.microbiomeanalyst.ca/)[89,90]. The SCFA data analysis from the stool fermentation was conducted for each SCFA and total SCFA production using one-way ANOVA followed by Tukey's post hoc test, with $P$ values adjusted for multiple comparisons using the software GraphPad Prism (v9.0, GraphPad Software LLC, San Diego, CA).

Separately, analysis of stool microbiome shotgun metagenome sequence data from PD participants was performed. Analyses of alpha- and beta-diversity were used to examine changes in microbial community structure between baseline and after the prebiotic intervention. Subgroup analysis of newly diagnosed, non-medicated, and treated PD participants were also performed. Alpha-diversity metrics (i.e., Shannon index, Simpson's index, richness, and evenness) were calculated from rarefied datasets (1 million sequences/sample) at the taxonomic level of species. Group comparisons were performed with a two-tailed, Wilcoxon signed-rank paired test in GraphPad Prism (v9.0, GraphPad Software LLC, San Diego, CA). Permutation Multivariate Analysis of Variance (PERMANOVA)[91] and Permutational Analysis of Multivariate Dispersions (PERMDISP)[92] were utilized to compare microbial community structure before and after the prebiotic intervention at the taxonomic level of species. The significance of PERMANOVA and PERMDISP values were determined using 9999 permutations and corrected using the Benjamini−Hochberg method. Visualization of baseline microbial community structure of newly diagnosed, non-medicated and treated PD participants was performed using principal coordinates analysis (PCoA) based on a Bray−Curtis dissimilarity distance matrix within the software package QIIME2[87]. Nonmetric multidimensional scaling (NMDS) plots of the bacterial species community were used to visualize data from baseline and after the prebiotic intervention. Each sample was connected to a centroid representing the mean of the group (i.e., baseline or after prebiotic). The Wilcoxon signed-rank test was used to identify significantly differentially abundant features (i.e., taxa, genes/pathways) between baseline and after the prebiotic intervention. Results were corrected using the Benjamini−Hochberg method. Differences in the relative abundance of individual taxa and functional genes/pathways with relative abundance greater than 0.1% were assessed for significance. Spearman's rank correlation coefficient was generated and corrected, using the Benjamini−Hochberg method, between the relative abundance of species, experimental measures, and clinical parameters. Data used for correlation analysis were transformed using Log2 fold change from baseline.

### Reporting summary

Further information on research design is available in the Nature Portfolio Reporting Summary linked to this article.

## Data availability

Individual participant data that underlie the results in this article after deidentification as well as analytic code will be made available immediately following publication indefinitely at the links provided below to anyone who wishes to access the data for any purpose. All sequencing reads generated in this study have been deposited in the National Center for Biotechnology Information (NCBI) BioProject database under accession numbers PRJNA852512 (16 S rRNA) and PRJNA756556 (Metagenomics). The SILVA 16 S rRNA database used for alignment is available at https://data.qiime2.org/2022.2/common/silva-138-99-515-806-nb-classifier.qza. The data and analyses generated in this study are available in two Source data files−16 S rRNA sequencing: https://github.com/ThaisaJungles/prebioticmixPD and shotgun metagenomics: https://zenodo.org/record/7327971. Source data are provided with this paper.

## Code availability

Custom code used to generate figures and statistical comparisons are available for 16 S rRNA sequencing data (https://github.com/ThaisaJungles/prebioticmixPD) and shotgun metagenomic data (https://zenodo.org/record/7327971).

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

## Acknowledgements

This study was funded by the Consolidated Anti-Aging Foundation, Christine and John Bakalar, an anonymous donor, NIH R24 AA026801 (A.K.), and NIH R01 AG056653 (R.M.V.). A.K. would like to acknowledge philanthropic funding from Mrs. Barbara and Mr. Larry Field, Mrs. Ellen and Mr. Philip Glass, Mrs. Marcia and Mr. Silas Keehn, and the Sklar Family. The sponsors had no role in study design, data collection and analysis, or manuscript writing. Prebiotic bars were provided by BetterBiotics Inc. The study team would also like to thank the study participants. The Rush Parkinson's Disease and Movement Disorder Program is a Parkinson's Foundation Clinical Center of Excellence.

## Author contributions

D.A.H.: study design, patient recruitment, patient assessment, data analysis/interpretation, and wrote the first draft of the manuscript; R.M.V.: conceptualization of the hypothesis and study, study design, data analysis, data interpretation, wrote, reviewed, and edited the manuscript, and revisions of the manuscript; T.M.C.J.: conceptualization of the prebiotic mixture, performed fermentation studies, performed stool SCFA measurements, data analysis and interpretation of the fermentation studies, review, and edit of the manuscript; B.H.: conceptualization of the prebiotic mixture, supervised fermentation studies, data analysis and interpretation of the fermentation studies, and review and edit of the manuscript; P.A.E.: processed biological samples, stored biological samples in repository, performed supervised microbiome interrogation, microbiome data analysis and interpretation, and review and edit of the manuscript; M.S.: supervised blood-based laboratory measurements, data analysis of blood-based laboratory measures, and review and edit of the manuscript; S.R.: performed blood-based laboratory measurements, data analysis of blood-based laboratory measures, and review and edit of the manuscript; T.C.: performed fermentation of PD stool with different fiber types, and review and edit of the manuscript; S.J.G.: supervised microbiome interrogation, microbiome data analysis and interpretation, conceptualization of the prebiotic bar, and review and edit of the manuscript; A.N.: performed microbiome bioinformatics analysis, review and edit of the manuscript; C.B.F.: study design, data analysis/interpretation, and review and edit of the manuscript; R.M.: data analysis/interpretation, and review and edit of the manuscript; B.O.: statistical analysis and review and edit of the manuscript; H.E.R.: conceptualization of the prebiotic bar, dietary analysis, and review and edit of the manuscript; S.S.: conceptualization of the prebiotic bar and review and edit of the manuscript; C.G.G.: study design, patient recruitment, patient assessment, data analysis/interpretation, and review and edit of the manuscript; A.K.: conceptualization of the hypothesis and the study, cowrote the first draft of the manuscript, conceptualization of the prebiotic mixture and the prebiotic bar; study design, data interpretation, and review and edit of the manuscript.

## Competing interests

The prebiotic bar was provided by BetterBiotics Inc which is co-owned by Drs. Keshavarzian, Hamaker, and Sedghi. A provisional patent is pending for the prebiotic mixture used in the bar (Patent Applicant: Purdue Research Foundation, Inventors: Drs. Hamaker, Cantu-Jungles, Keshavarzian, Application Number: 69548-02, Status: Pending, Patent covers the use of prebiotics to improve health). Drs. Keshavarzian, Hamaker, Sedghi, and Cantu-Jungles were not involved in subject recruitment, subject assessments, or statistical analysis of the data. The remaining authors declare no competing interests.
