## [Peer Review File · Nature Communications]

An Open Label, Non-Randomized Study Assessing a Prebiotic Fiber Intervention in a Small Cohort of Parkinson's Disease ParticipantsReviewers' Comments:

Reviewer #1:

Remarks to the Author:

This is a great proof-of-concept study that supports the notion of targeting gut-derived processes (in this case, fiber fermentation to generate beneficial SCFA) to modulate Parkinson's outcomes in a patient cohort.

The data are very supportive (even given the low N), and certainly warrant stimulating increased thinking in the field. As such, this reviewer only has a few minor concerns that would be worthwhile to consider/address prior to publication. These are presented generally in order of appearance.

pg 5, line 99- The authors state: "Although there is no unique microbiota signature for PD." This reviewer would disagree a bit. It is difficult to claim uniqueness (even among neurodegenerative diseases), since not all have been as well characterized as the PD-associated microbiome. However, this reviewer would argue that there is a signature appearing across the published datasets. When looking at all data holistically, a signature of increased Akkermansia, Lactobacillus, Bifidobacteria, Enterobacteriaceae and a decrease in Lachnospiraceae appears. Certainly, some discrepancies as to the species identification within these genera/families, and the magnitude of the differences. But these, and a few others, really do start to bubble to the surface as a microbiome signature of PD.

The authors show that GI symptoms are reduced in treated PD patients, but not de novo, untreated patients. There is some discussion here on pg 20 regarding this. However, this could be expanded a bit. Perhaps direct analysis of the pre-treatment microbiome between the two different patient groups to identify particularly sensitive taxa? It seems that in the De novo group, that more taxa changed following fiber treatment. Some more clarity as to why L-dopa treated patients might respond more therapeutically would be useful.

Given that the majority of microbiome alterations for SCFA producers observed are those which correlated with age and treatment duration, is it possible that these are not necessarily a factor in the etiology- but rather a consequence of the disease or treatment? Is it possible to begin assign directionality? This should be a clarified a touch.

Given the barrier dysfunction, is it possible that those with PD longer (the treated group) have a more permeable barrier (is zonulin comprehensive for permeability?) that allows more SCFA to cross during treatment? This may explain the increased SCFA in plasma of the treated PD cohort.

Some minor proof reading is needed, for instance- E. Coli has the species incorrectly capitalized throughout.

Overall, this is a clear and straightforward study that certainly does what the authors hope "provide scientific rationale for a microbiota-directed prebiotic" for PD.

Reviewer #2:

Remarks to the Author:

This proof of concept study by Hall and coworkers addresses an important neurodegenerative disorder for which there is no medical treatment to stop the disease process or the deterioration of quality of life. Any evidence-based complementary strategy that improves quality of life, physically or mentally, deserves attention and is of significance to the field. Hall and coworkers demonstrate a relatively rapid effect on PD-associated motor symptoms after consumption of a prebiotic fiber mixture. They propose a mechanism involving SCFA as main mode of action. The findings are of interest albeit the group size (n=10/group) and the apparent lack of a placebo control bar are no methodological strengths as the authors realized themselves. As detailed below, it is necessary to provide more information about the

specificity and sensitivity of specific analytical methods (SCFA and some of the biomarkers). Suggestive text passages related to metrics that are merely proxy readouts have to be supported by additional data or deleted. Comment 6. would help to address concerns regarding the design as discussed by the authors themselves.

1. The authors should provide additional information about the sensitivity of their analytical SCFA method. While the analysis of samples from in vitro studies or fecal samples are usually relatively straight forward, accurate quantification of SCFA in plasma/serum is more difficult because of other fatty acids. The additional (supplementary) information should clarify how reliable the authors were able to discriminate the three SCFA (especially acetate) from other short/medium-chain fatty acids (e.g. BCFA). Overall, the reported concentrations prior to treatment are consistent with plasma SCFA concentrations measured in clinical studies.

2. LPS-binding protein is a liver-derived class 1 acute phase protein. Its transcriptional regulation is similar to other APP (AP1, C/EBP etc). Why should the induction of this molecule be specific for bacterial LPS? What is the exact experimental evidence to designate this molecule "a marker of bacterial translocation"? The name of this molecule is suggestive and implies a specificity which is in my view not backed-up experimentally.

3. The specificity and sensitivity of the assay for zonulin from MyBioSource is debatable. The authors should provide information about specificity, including spike retrieval data. At which dilution was the actual assay performed? Calibration curve values should be provided together with OD data of the actual samples (supplementary data).

4. Suggestive passages in the Result section like ("indicating that the prebiotic intervention significantly improved intestinal barrier integrity") should be deleted. This writing style may unintendedly lead to suggestive overstatements as in lines 79-80 of the Abstract. The biomarkers and bacteria measured herein are at most markers that have been associated with 'intestinal barrier integrity and intestinal inflammation'. Intestinal barrier integrity and intestinal inflammation were not directly assessed, merely proxy's thereof. Such claims can be made once functional tests have been performed in a subsequent trial, and intestinal inflammation has been evaluated e.g. in a parallel animal study or once the mixture is tested as a treatment prior to bariatric surgery.

5. The sensitivity of the MSD platform for NF-L is rather modest and more detailed information should be provided about the raw data, i.e. whether samples were determined at the lower end of the calibration curve (same as 4.). Is there a correlation between age and NF-L?

6. The proposed SCFA-related mechanism is not necessarily the main mode of action of the treatment. Support for a predominantly SCFA-related mechanism requires additional in vitro fermentation experiments using feces from PD patients – ideally using feces from de novo and levodopa treated patients. One would expect reduced SCFA production rates relative to healthy controls, at least during the first 12-24 hours of the in vitro experiment. On the other hand, it is also possible that the microbiota of PD patients is not impaired regarding its capacity to produce SCFA. The latter would support the idea that availability of substrate rather than 'dysbiosis' per se is critical (although this would not justify to omit a placebo control in future experiment).

7. The conclusions about intestinal inflammation are based on modulation of bacterial species and thus an overstatement. The effects seem to be metabolic in nature and associated anti-inflammatory effects are typically at tissue level (not investigated herein). Such effects are often not reflected systemically and measured cytokines are probably not derived from the tissues exposed to highest fluxes of SCFA. Respective passages should be rigorously rephrased to avoid oversimplifications.

8. Given the importance to patients suffering from PD, the etiological heterogeneity of the disease patients and the possible conclusions that will be drawn from any proof of concept study, one would expect larger groups, even for PoC studies. What were the main arguments to choose n=10/group? The levodopa treatment median dose in table 1 should be substituted by a dose range which would be more meaningful.

Reviewer #3:

Remarks to the Author:

Interesting findings on zonulin and fecal calprotectin

The microbiome data lacks a lot of data

for example what races were the patients more specific than hispanic or black, the origin of patients is very important as different regions of the planet exhibit different increases in relative abundance of certain microbes. Age and diet also matters when demonstrating the microbiome of 10 patients.

Work is original but the data on the microbiome cannot be validated and is not reproducible.

Was proteobacteria phylum present in all parkinson's patients? what about bifidobacteria or F prausnitzii. ?

As you know it is hard to find markers in Parkinson's if everyone is different but more importantly it is hard to make a case that prebiotics helps these microbes if assay in itself is not validated, verified and reproducible. The data on fecal calprotectin and zonulin is impressive but not the microbiome data.

Reviewer #4:

Remarks to the Author:

Dear Editor,

It was a pleasure to review the manuscript of Hall et al describing the results of a proof-of-concept open-label, non-randomized study in de novo and treated Parkinson's disease patients. The authors examined the effects of a prebiotic mixture on a comprehensive set of features in a relative small group of patients (20 patients in total and studied its effect on gut microbiota composition, faecal levels of short-chain fatty acids and a marker of intestinal inflammation, blood markers of inflammation, intestinal barrier integrity and neuronal health, as well as on gastrointestinal and Parkinson's motor symptoms, using a before/after comparison.

Although the study design warrants caution (no controls, no blinding, small sample size), the authors report several noteworthy results: prebiotic intervention was tolerable in patients and resulted in improvement of gastrointestinal symptoms, changes in the abundances of several SCFA-producing gut bacterial taxa, increased faecal SCFA levels, and reduction of plasma markers of intestinal barrier dysfunction and neurodegeneration. The laboratory methods seem appropriate for the chosen features.

The within-subject design of the study and statistical analyses and complementation of in vivo findings with in vitro experimental work are relative strengths. Especially the finding that the different prebiotic components support different bacterial species and functions is highly relevant. The manuscript overall is well written and has a good, logical structure.

Some major remarks;

- The prebiotic was administered for a relative short period of time. Despite, the authors did find significant changes in microbes, metabolites, inflammation etc. after only 10 days. To me that is slightly surprising. Was this to be expected? Can this be potentially explained by confounding effects (i.e. did patients change their diet? Were they advised to change/alter/optimize their diet? Is this possibly a placebo effect?)
- Some effects were different for de novo patients versus and treated patients. On page 21 the authors discuss that "prebiotics may be especially beneficial for PD patients treated by levodopa". Could you clarify if in this context you are referring only to treatment status, or is time from disease onset or severity also important to consider?
- The authors describe that environmental factors that are known risk factors for Parkinson's disease also affect the gut microbiome. Though that would strengthen the claim the authors make on page 5 line 91 that "it is plausible that the gut mediates effects of environmental PD risk". Though important, I think it is less relevant here, and might be even confusing, as authors aim with these studies to not prevent development of PD, but to ameliorate/alter symptoms once diagnosed with the disease.
- There is (too) little information on the methodological side of things, for example information on number of repeat experiments (in vitro), controls (in vitro), some clinically relevant information, methods for work up of faecal samples, and data analyses of in vivo data etc. (for details and suggestions; see below). It is extremely important to provide such information to value the robustness

of results, and to be able to reconstruct these experiments.

- It has to be noted that 3 of the authors are owner of the company that had produced the prebiotic bar, and/or have submitted a provisional patent for the prebiotic mixture used in the bar, though the conflict of interest statement mentions those authors were not involved in subject recruitment, subject assessments, or statistical analysis of the data, showing they have tried to minimise the potential influence of this on study results.
- For the benefit of their patients, because of these promising results, I would strongly urge the authors to initiate a proper randomized controlled double blind trial to prove the efficacy of the product with high certainty. This should be feasible as authors already showed that only a relative low number of patients is needed to reach significance.
- Given the importance of adherence, I would advise the authors to provide the evidence base for information (questionnaire?) on adherence, palatability etc. if available. It is not mentioned in the methods either.

Below further detailed comments:

Abstract:

- As mentioned, one of the strengths of this study is the use of an in vitro model alongside clinical sampling, thus, it is worth mentioning the main results from the in vitro study in the abstract as well.
- A small grammatical error on page 4 lines 77-78: "while relative abundance short-chain-fatty acids-producing bacteria were increased" should be change to "while the relative abundance of short-chain-fatty acid-producing bacteria was increased"

Introduction:

- The aims of the study should also list the in vitro study.
- There is a repetition of "SCFA-producing" in the sentence listing the aim 2 on page 6 line 122

Methods:

- In vitro study:
 - o There is a repetition of "slow fermentation" in the sentence on page 7 line 133-134. Could you clarify what it means in the context that they are both tolerable and allow to be delivered to the distal parts?
 - o Could you clarify in how many replicates the experiments were conducted? Are there independent biological replicates or only technical replicates?
 - o Did you include any controls in the experiments? In Figure 1 there is a mention of a blank, but it is not described in the Methods.
 - o Could you provide information for the bioinformatic processing of the 16S rRNA sequencing, including pipelines, decontamination, data cleaning procedures?
 - o Please also provide information on the statistical analyses of the relative abundances. Were these conducted compared to the control/blank group?
- Clinical study:
 - o Could you clarify if it was specified to the patients how and when they had to eat the prebiotic bar? I.e. what time of day, with food or not?
 - o Could you specify the faecal sample input amount to the DNA extraction protocol?
 - o In the paragraph on page 13 lines 273-276 could you specify if these analyses were also performed within-subjects, i.e. comparing data at baseline and post-prebiotic intervention?
 - o In regards of multiple comparison correction, the authors say on page 13 line 272 that "No adjustment for multiple testing was applied to this pilot study". Does this apply only to the demographic/clinical data or also for the analysis of the blood and SCFA markers?
 - o Page 13 lines 284-286: there seem to be some words missing in the sentence. Do you mean that the NMDS plot was used to visualise the bacterial community structure of the samples?
 - o Could you correct the sentence on page 13 line 287 to page 14 line 290? The Wilcoxon-signed rank test is a statistical hypothesis test and not a method for multiple comparison correction. What method did you use for adjusting for multiple comparisons here? Additionally, "matched samples" might be a better alternative to "paired analyses" in this sentence.
 - o Analyses relating to differential relative abundances: please specify if you performed comparisons for all taxa identified in samples or only the top taxa presented in Fig 2c?

o What methods did you use to evaluate the outcome data distribution and homogeneity of variances?
Results, figures and tables:

- Could you explain the motivation for the correlation analyses starting from page 18 line 396?

Although these results are interesting, currently it remains unclear what you were hoping to achieve by performing these analyses. In addition, could you explain if the correlation analyses were performed at baseline, post-prebiotic or taking all pre- and post-treatment samples into consideration?

- Figure 1:

o page 24 lines 521-522: the authors say that "Bars denoted by a different letter indicate significant differences between treatment means ($p < 0.05$)", however, it is not specified what each of the different letter mean.

o Please mention in the legend what the error bars represent on Fig 1b-e.

- Figure 2:

o Line 545: in "Non-metric MDS" – explain the abbreviation in the legend

o Differentially abundant bacteria before and after prebiotics: are these analyses adjusted for multiple comparisons? Given the number of comparisons performed, even as a proof-of-concept study, it would be crucial to correct for multiple testing. It appears from Table S1 that some adjustments for multiple testing was performed, but this should be made explicit in the Methods and main text figures.

o e-j: what was the basis of choosing to show individual plots of these 6 taxa and leave out two other that had differential abundance (*Ruminococcus torques* and *bromii*)?

o It is a strength that individual datapoints are shown in d-k.

- Table 2:

o I do not understand the values and IQR values provided for the tolerability data. It is a yes/no answer, correct, or a scale? As mentioned before, insights into the questions asked is relevant.

- Table S1:

o It might be better not to report p- and q-values mixed in the last column. As an alternative, the authors could report both the unadjusted p-values and FDR-corrected p-values (q-values) side by side in two columns as done for Table S2.

Discussion:

- Page 22 line 486, sentence starting with "However, this is less likely because ..." – the meaning of this sentence is unclear. There is also a double negative ("not non-fermenting").

- Page 22 line 490: what does it mean that samples were coded? Do you mean the laboratory work was performed blinded to the timepoint and patient group?

- A major strength in the current study is the within-subjects design (i.e. always comparing baseline and post-prebiotic data of the same participant) and thus could be mentioned in the limitations/strength paragraph of the Discussion.

Response to Referees

Point by point responses are found below our responses are in **green** and changes in the manuscript are indicated in **blue**.

Reviewer 1

This is a great proof-of-concept study that supports the notion of targeting gut-derived processes (in this case, fiber fermentation to generate beneficial SCFA) to modulate Parkinson's outcomes in a patient cohort. The data are very supportive (even given the low N), and certainly warrant stimulating increased thinking in the field. As such, this reviewer only has a few minor concerns that would be worthwhile to consider/address prior to publication. These are presented generally in order of appearance.

Response: *Thank you for noting that the data are supportive of the notion of targeting gut-derived processes in PD and warrant increased thinking in field.*

(Comment 1) pg 5, line 99- The authors state: "Although there is no unique microbiota signature for PD." This reviewer would disagree a bit. It is difficult to claim uniqueness (even among neurodegenerative diseases), since not all have been as well characterized as the PD-associated microbiome. However, this reviewer would argue that there is a signature appearing across the published datasets. When looking at all data holistically, a signature of increased Akkermansia, Lactobacillus, Bifidobacteria, Enterobacteriaceae and a decrease in Lachnospiraceae appears. Certainly, some discrepancies as to the species identification within these genera/families, and the magnitude of the differences. But these, and a few others, really do start to bubble to the surface as a microbiome signature of PD.

Response: *Thank for noting this important nuance. While there is no specific signature for PD, as you rightly point out, there is a commonly observed pattern including increased relative abundance of pathobionts, decreased relative abundance of short chain fatty acid (SCFA)-producing bacteria, as well as consistent changes in specific individual taxa (e.g., Akkermansia, Lactobacillus, Bifidobacteria). Accordantly, we have revised the statement in question to emphasize commonly associated features of the PD microbiota (line 99).*

(Comment 2) The authors show that GI symptoms are reduced in treated PD patients, but not de novo, untreated patients. There is some discussion here on pg 20 regarding this. However, this could be expanded a bit. Perhaps direct analysis of the pre-treatment microbiome between the two different patient groups to identify particularly sensitive taxa? It seems that in the De novo group, that more taxa changed following fiber treatment. Some more clarity as to why L-dopa treated patients might respond more therapeutically would be useful.

Response: *Indeed, prebiotic treatment had beneficial effects on gastrointestinal symptoms in treated PD participants while no obvious impact was noted in de novo PD participants. Our analysis showed no differences in microbiota communities at baseline between de novo and treated PD participants that could account for the differences; nor were there differences between these groups after the prebiotic intervention (these data are now included in the revised manuscript: Table S2, Figure S1). We posit that the factor underlying differences in GI symptoms in de novo and treated PD participants is related to the severity of symptoms of each group at baseline. Treated PD patients had a higher total GI symptom score at baseline than de novo patients suggesting that the greater the degree of GI symptoms the greater the benefit of the prebiotic intervention. This point is now included in the revised manuscript (line 418).*

(Comment 3) Given that the majority of microbiome alterations for SCFA producers observed are those which correlated with age and treatment duration, is it possible that these are not necessarily a factor in the etiology- but rather a consequence of the disease or treatment? Is it possible to begin assign directionality? This should be a clarified a touch.

Response: *We agree that the observed changes in the microbiota community do not indicate a causal link between PD etiology and the microbiota and it is possible that at least some of the observed changes in microbiota community are a consequence of age, PD, or PD treatment. For example, in this study Eubacterium siraeum and Ruminococcus bircirculans correlated with age and Bifidobacterium adolescentis correlated with levodopa daily dose. Certainly, medication use is well-established to modify the intestinal microbiota,^{1, 2} but in this study no significant differences were noted in microbiota communities*

in de novo and treated PD patients at baseline (new Figure S2). Unfortunately, it is not possible to determine if the altered microbiome drives or is a consequence of PD or treatment in this study. Studies spanning decades will be required to delve into this incredibly important question.

(Comment 4) Given the barrier dysfunction, is it possible that those with PD longer (the treated group) have a more permeable barrier (is zonulin comprehensive for permeability?) that allows more SCFA to cross during treatment? This may explain the increased SCFA in plasma of the treated PD cohort.

Response: *In the current study, PD treatment duration was not correlated with serum zonulin or LBP ($p>0.05$) suggesting that treatment duration likely does not influence the barrier. However, this manuscript as well as previous publications by our group and others demonstrate that PD patients have intestinal barrier dysfunction using direct methods including a urinary sugar test and ex vivo analysis of intestinal tissue (immunohistochemistry of tight junction proteins in intestinal biopsy tissue) as well as indirect assessments such as LBP, LPS, and zonulin.³⁻⁸ Approximately 90-95% of SCFA are absorbed in the colon via a process involving passive diffusion or ion exchange,^{13, 14} suggesting the majority of SCFA are absorbed. However, a recent publication demonstrates that serum SCFA levels may be influenced by intestinal barrier integrity wherein SCFA levels are correlated with a marker of barrier integrity (serum ZO-1).¹⁵ Additional studies are needed to fully understand if SCFA absorption is influenced by the intestinal barrier in PD patients.*

(Comment 5) Some minor proof reading is needed, for instance- E. Coli has the species incorrectly capitalized throughout.

Response: *Thank you for noting this oversight. The revised manuscript has been carefully proofed and edited accordingly.*

Reviewer #2

This proof of concept study by Hall and coworkers addresses an important neurodegenerative disorder for which there is no medical treatment to stop the disease process or the deterioration of quality of life. Any evidence-based complementary strategy that improves quality of life, physically or mentally, deserves attention and is of significance to the field. Hall and coworkers demonstrate a relatively rapid effect on PD-associated motor symptoms after consumption of a prebiotic fiber mixture. They propose a mechanism involving SCFA as main mode of action. The findings are of interest albeit the group size ($n=10$ /group) and the apparent lack of a placebo control bar are no methodological strengths as the authors realized themselves. As detailed below, it is necessary to provide more information about the specificity and sensitivity of specific analytical methods (SCFA and some of the biomarkers). Suggestive text passages related to metrics that are merely proxy readouts have to be supported by additional data or deleted. Comment 6. would help to address concerns regarding the design as discussed by the authors themselves.

Response: *Thank you for noting that an evidence-based strategy to improve quality of life deserves attention and is of significance. Additional methodological details have been added (e.g., sensitivity, analytical methods) to the revised manuscript to ensure rigor and reproducibility. Finally, suggestive text has been omitted throughout the manuscript. The response to Comment 6 is detailed below.*

(Comment 1) The authors should provide additional information about the sensitivity of their analytical SCFA method. While the analysis of samples from in vitro studies or fecal samples are usually relatively straight forward, accurate quantification of SCFA in plasma/serum is more difficult because of other fatty acids. The additional (supplementary) information should clarify how reliable the authors were able to discriminate the three SCFA (especially acetate) from other short/medium-chain fatty acids (e.g. BCFA). Overall, the reported concentrations prior to treatment are consistent with plasma SCFA concentrations measured in clinical studies.

Response: *Thank you for raising the important issue of quantification of SCFA and noting that the levels reported in our study are consistent with those found in other studies. Retention times were used to differentiate acetate, butyrate, propionate from other metabolites (e.g., branch-chain fatty acids, isobutyric acid, isovaleric acid, isopropionate). The limit of detection (LOD) was 0.3-0.6ug/ml for acetate and 0.03-0.12ug/ml for propionate and butyrate. This important information is now included in the Methods section of the revised manuscript (line 270) and we cite a recent publication by our group which uses and describes these methods.¹⁶*

(Comment 2) LPS-binding protein is a liver-derived class 1 acute phase protein. Its transcriptional regulation is similar to other APP (AP1, C/EBP etc). Why should the induction of this molecule be specific for bacterial LPS? What is the exact experimental evidence to designate this molecule “a marker of bacterial translocation” The name of this molecule is suggestive and implies a specificity which is in my view not backed-up experimentally.

Response: *The inflammatory response to LPS is mediated via LPS-binding protein (LBP) wherein LPS binds to LBP to form a complex that subsequently binds to CD14 to elicit an immune response.¹⁷⁻¹⁹ The levels of LBP in serum are increased during sepsis and LBP levels are upregulated in disorders/diseases associated with chronic, low grade inflammation induced by microbiota dysbiosis and intestinal barrier dysfunction²⁰⁻²⁴ including PD.³⁻⁶ However, we agree that LBP may be a non-specific marker for inflammation rather than specific for bacterial translocation. To this end, we measured serum C reactive protein (CRP, another liver-derived acute phase protein) and serum cytokines (markers of systemic inflammation). The prebiotic intervention did not impact CRP or serum cytokines. Taken together this data suggests that 10 days of prebiotic intervention that changes the microbiota and intestinal inflammation (calprotectin) is not sufficient to alter systemic inflammation.*

(Comment 3) The specificity and sensitivity of the assay for zonulin from MyBioSource is debatable. The authors should provide information about specificity, including spike retrieval data. At which dilution was the actual assay performed? Calibration curve values should be provided together with OD data of the actual samples (supplementary data).

Response: *Thank you for this suggestion. The revised manuscript includes an expanded methods section and now detection ranges (standard range, level of detection) and dilutions (when relevant) for all assays conducted. Specifically, the detection range for the zonulin assay is 0.625 – 40 ng/ml and the sensitivity is <0.156ng/ml, the spike retrieval data average percent recovery was 97% in plasma (range: 90-104). Additional information has been provided for all assays in this study including the range of detection, dilution (if applicable), it is noted if samples were run in duplicate or triplicate, and if it is noted if data fell within the range of detection (**Methods** section). We hope this additional information lends confidence in the results not only for the zonulin assay but also for all outcomes assessed in this study.*

(Comment 4) Suggestive passages in the Result section like (“indicating that the prebiotic intervention significantly improved intestinal barrier integrity”) should be deleted. This writing style may unintendedly lead to suggestive overstatements as in lines 79-80 of the Abstract. The biomarkers and bacteria measured herein are at most markers that have been associated with “intestinal barrier integrity and intestinal inflammation”. Intestinal barrier integrity and intestinal inflammation were not directly assessed, merely proxy’s thereof. Such claims can be made once functional tests have been performed in a subsequent trial, and intestinal inflammation has been evaluated e.g. in a parallel animal study or once the mixture is tested as a treatment prior to bariatric surgery.

Response: *Thank you for noting this important point. We agree that serum zonulin, and LBP are, at best, only markers of intestinal barrier integrity and bacterial translocation. Studies demonstrate that zonulin levels are increased in patients with chronic inflammatory diseases known to have disrupted intestinal barrier integrity;⁹⁻¹² however, we acknowledge the controversy regarding the use of serum zonulin as a marker of barrier integrity. The current study demonstrated that serum zonulin was reduced by the prebiotic intervention, but we cannot conclusively state that intestinal barrier dysfunction was improved by the prebiotic intervention. Accordingly, we have softened the interpretation that zonulin represents barrier integrity and highlight the need for future studies using more direct assessments of the intestinal barrier to confirm that a prebiotic intervention improves barrier integrity in PD patients (**line 567**). However, intestinal inflammation was directly assessed in this study using stool calprotectin. Calprotectin is both a reliable and specific marker of intestinal inflammation. Stool calprotectin is elevated in disorders associated with intestinal inflammation like inflammatory bowel disease (IBD) and levels of stool calprotectin correlate with severity of intestinal inflammation and decreases after treatment with biologics.²⁵ Indeed, the use of stool calprotectin to monitor intestinal inflammation and assess response to treatment in patients with IBD is a cornerstone of management of patients with IBD.²⁶ This study assessed stool calprotectin and found that PD patients have intestinal inflammation and the prebiotic intervention decreased intestinal inflammation in PD patients. This important information is now included in the revised manuscript (**line 275**).*

(Comment 5) The sensitivity of the MSD platform for NF-L is rather modest and more detailed information should be provided about the raw data, i.e. whether samples were determined at the lower end of the calibration curve (same as 4.). Is there a correlation between age and NF-L?

Response: *The R-PLEX Human Neurofilament L Antibody Set was used for analysis. The calibration curve was fitted with a 4-parameter logistic model with a 1/Y² weighting. The LLOD is a calculated concentration corresponding to the signal 2.5 standard deviations above the background zero calibrator. Detection Range: 5.5-50,000 pg/ml. The samples averaged 70.02 at baseline and 63.15pg/ml after prebiotic treatment (all samples were within range of the assay, but at the lower end of the detection range). This information is now included in the Methods section of the revised manuscript. Analysis of the baseline data shows that there is a significant positive correlation between age and NfL ($p = 0.03$, $r = 0.50$, $R^2 = 0.25$); however, the prebiotic intervention decreased serum NfL and our conclusion that these levels are decreased by the prebiotic intervention is not impacted by age. Additional information about the NfL assay is now included in the revised manuscript (line 312).*

(Comment 6) The proposed SCFA-related mechanism is not necessarily the main mode of action of the treatment. Support for a predominantly SCFA-related mechanism requires additional in vitro fermentation experiments using feces from PD patients, ideally using feces from de novo and levodopa treated patients. One would expect reduced SCFA production rates relative to healthy controls, at least during the first 12-24 hours of the in vitro experiment. On the other hand, it is also possible that the microbiota of PD patients is not impaired regarding its capacity to produce SCFA. The latter would support the idea that availability of substrate rather than “dysbiosis” per se is critical (although this would not justify to omit a placebo control in future experiment).

Response: *Thank you for this insightful comment. To support the SCFA-mechanism we have included additional data in the revised manuscript. We conducted an ex vivo fermentation study using stool from individuals with PD and age matched controls with multiple fibers including SCFA-promoting fibers. This study revealed that: (1) total SCFAs are lower in stool from PD patients compared to controls and (2) fermentation of PD stool with SCFA-promoting fibers significantly increase SCFA levels in PD stool comparable to SCFA levels found in control stool. These important new data which support SCFA as a potentially important mechanism have been added to the Results section of the revised manuscript (Figure S1, line 393). Nonetheless, we agree with the reviewer that SCFA are not necessarily the only mechanism, and several mechanisms are likely important in mediating the effect of microbiota manipulation on inflammation and the brain. This point has been addressed in the Discussion section of the revised manuscript (line 584).*

(Comment 7) The conclusions about intestinal inflammation are based on modulation of bacterial species and thus an overstatement. The effects seem to be metabolic in nature and associated anti-inflammatory effects are typically at tissue level (not investigated herein). Such effects are often not reflected systemically and measured cytokines are probably not derived from the tissues exposed to highest fluxes of SCFA. Respective passages should be rigorously rephrased to avoid oversimplifications.

Response: *It is true that information can be inferred about intestinal inflammation based on bacterial species, but in this study intestinal inflammation was directly assessed using calprotectin. Calprotectin is produced by neutrophils in the intestinal mucosa and stool calprotectin is a reliable, and widely used marker to assess intestinal inflammation including in Parkinson’s disease.^{7, 8, 27} Moreover, studies in patients with inflammatory bowel disease (IBD) show a direct correlation between calprotectin levels with intestinal inflammation assessed via endoscopy, histology, which can be changed by an intervention.²⁵ Additional references have been included in the revised manuscript to clarify this point (line 275). It is true that results from this study indicate that systemic inflammation was not impacted by the prebiotic intervention. The text has been revised to avoid oversimplifications as suggested.*

(Comment 8) Given the importance to patients suffering from PD, the etiological heterogeneity of the disease patients and the possible conclusions that will be drawn from any proof of concept study, one would expect larger groups, even for PoC studies. What were the main arguments to choose n=10/group? The levodopa treatment median dose in table 1 should be substituted by a dose range which would be more meaningful.

Response: *This proof-of-concept study was primarily designed to assess tolerability and acceptability of the prebiotic intervention in PD patients and determine whether a prebiotic intervention can modulate the microbiota community (n=20 total: n=10 de novo and n=10 treated). We selected a sample size of 20 to be*

conservative. If the prebiotic intervention did not impact the microbiota community in a small sample size, then its potential clinical impact would have been limited. This study provides the required feasibility to demonstrate that the prebiotic intervention influences the microbiota as predicted and impacts meaningful biological outcomes. This also guides a timeframe required for future studies wherein a longer treatment period may be required to observe changes in systemic inflammation. We agree that additional details about levodopa treatment are critically important. Accordingly, levodopa treatment dose including median and range) are now included in Table 1 of the revised manuscript.

Reviewer #3

Interesting findings on zonulin and fecal calprotectin

Response: Thank you for noting these strengths.

(Comment 1) The microbiome data lacks a lot of data for example what races were the patients more specific than hispanic or black, the origin of patients is very important as different regions of the planet exhibit different increase in relative abundance of certain microbes. Age and diet also matters when demonstrating the microbiome of 10 patients.

Response: All subjects were recruited from the Chicagoland area and this information is now included in the revised manuscript to accompany information about subject demographics including age, race (100% white) and ethnicity (100% not Hispanic) (Table 1). We agree that diet is critically important for any study examining the intestinal microbiota. Diet quality was assessed at baseline using the validated, 14-item Mediterranean Diet Adherence Screener (MEDAS)²⁸ and participants are asked to continue their usual diet for the duration of the 10-day study. Participants were moderately accordant to a Mediterranean diet (median 8.5, IQR 4.3). No differences in accordant existed between de novo and treated participants ($p = 0.96$). This additional information is now included in the revised manuscript (line 223, line 404).

(Comment 2) Work is original but the data on the microbiome cannot be validated and is not reproducible.

Response: Thank you for noting the originality of the work. Additional information about the analysis approaches have been added to the revised manuscript which will enable all readers to reproduce the experimental conditions using other cohorts (see revised Methods section). Additionally, the 16S rRNA sequencing data has been made publicly available and can be accessed on NCBI SRA PRJNA852512 (this information is now included in the revised manuscript).

(Comment 3) Was proteobacteria phylum present in all parkinson's patients? what about bifidobacteria or F prausnitizii?

Response: Thank you for these important questions. The revised manuscript now contains relative abundance ranges at baseline and after the prebiotic intervention for taxa of importance (line 440). Bacteria from the phylum Proteobacteria were present in all PD participants at baseline with a relative abundance range of 0.02-16.29%. Following the prebiotic intervention, three PD participants no longer had detectable Proteobacteria, with the relative abundance ranging from 0.00-6.99%. Similarly, *Bifidobacterium adolescentis* went from an average relative abundance from 0-40.12% (eight participants with 0.00%) at baseline to 0-54.62% (eight participants with 0.00%) after the prebiotic intervention. *Faecalibacterium prausnitizii* went from a relative abundance of 0-19.50% at baseline (one participant with 0.00%) to 0-24.57% after the prebiotic intervention. This information is now included in the revised manuscript.

Reviewer #4

Dear Editor, It was a pleasure to review the manuscript of Hall et al describing the results of a proof-of-concept open-label, non-randomized study in de novo and treated Parkinson's disease patients. The authors examined the effects of a prebiotic mixture on a comprehensive set of features in a relative small group of patients (20 patients in total and studied its effect on gut microbiota composition, faecal levels of short-chain fatty acids and a marker of intestinal inflammation, blood markers of inflammation, intestinal barrier integrity and neuronal health, as well as on gastrointestinal and Parkinson's motor symptoms, using a before/after comparison. Although the study design warrants caution (no controls, no blinding, small sample size), the authors report several noteworthy results: prebiotic intervention was tolerable in patients and resulted in improvement of gastrointestinal symptoms, changes in the abundances of several SCFA-producing gut bacterial taxa, increased faecal SCFA levels, and reduction of plasma markers of intestinal barrier dysfunction and

neurodegeneration. The laboratory methods seem appropriate for the chosen features. The within-subject design of the study and statistical analyses and complementation of in vivo findings with in vitro experimental work are relative strengths. Especially the finding that the different prebiotic components support different bacterial species and functions is highly relevant. The manuscript overall is well written and has a good, logical structure. Some major remarks;

Response: *Thank you for highlighting the noteworthy findings and noting the within-subject design and statistical analyses as strengths.*

(Comment 1) The prebiotic was administered for a relative short period of time. Despite, the authors did find significant changes in microbes, metabolites, inflammation etc. after only 10 days. To me that is slightly surprising. Was this to be expected? Can this be potentially explained by confounding effects (i.e. did patients change their diet? Were they advised to change/alter/optimize their diet? Is this possibly a placebo effect?)

Response: *Thank you for this important question. It is well-established that microbiota composition is rapidly impacted by diet;²⁹ therefore, the changes in microbiota community observed over the 10-day prebiotic intervention are not unexpected. We agree that we did not expect such significant changes in other biological markers after only 10 days of treatment. However, the immune system can rapidly respond to changes in the microbiota, while inflammation-driven biological factors may take longer to respond. Participants were advised to continue habitual diet, activity, and sleep during the 10-day study, but we cannot discount the possibility of a placebo effect and look forward to conducting placebo-controlled trials in the future to address this important point. This point is now included in the revised manuscript (line 589).*

(Comment 2) Some effects were different for de novo patients versus and treated patients. On page 21 the authors discuss that prebiotics may be especially beneficial for PD patients treated by levodopa. Could you clarify if in this context you are referring only to treatment status, or is time from disease onset or severity also important to consider?

Response: *Thank you for this question. There were no significant differences between microbiota communities in de novo and treated groups at baseline (new Table S3, Figure S2), but there are a plethora of data showing that medication use influences the intestinal microbiome. A recent publication by our group indicate that levodopa dose is impacted by the microbiota community wherein those with higher abundance of bacteria containing Tyrosine decarboxylase (TDC) require a higher daily levodopa dose to see therapeutic benefits.³⁰ The statement in question refers to treatment status (de novo vs treated) as future studies with larger treatment groups are needed to delve into the important questions relating to age at disease onset and severity of symptoms.*

(Comment 3) The authors describe that environmental factors that are known risk factors for Parkinson's disease also affect the gut microbiome. Though that would strengthen the claim the authors make on page 5 line 91 that "it is plausible that the gut mediates effects of environmental PD risk". Though important, I think it is less relevant here, and might be even confusing, as authors aim with these studies to not prevent development of PD, but to ameliorate/alter symptoms once diagnosed with the disease.

Response: *Thank you for this suggestion. Indeed, this proof-of-concept study is not focused on risk of PD but instead on how the feasibility / acceptability of the prebiotic intervention and how prebiotic-induced microbiota modification impacts those with PD. The statement in question has been revised to better align with the goals of this study (line 93).*

(Comment 4) There is (too) little information on the methodological site of things, for example information on number of repeat experiments (in vitro), controls (in vitro), some clinically relevant information, methods for work up of faecal samples, and data analyses of in vivo data etc. (for details and suggestions; see below). It is extremely important to provide such information to value the robustness of results, and to be able to reconstruct these experiments.

Response: *We too see the value of including these details. Experimental details have been expanded in the revised manuscript including use of controls, assay ranges (lowest level of detection), technical duplicates (if samples were run in duplicate/triplicate), if data were within the range of the assay, and raw sequence data were deposited in the National Center for Biotechnology Information (NCBI) Sequence Read Archive, under the identifier PRJNA852512. This information is found throughout the Methods section.*

(Comment 5) It has to be noted that 3 of the authors are owner of the company that had produced the prebiotic bar, and/or have submitted a provisional patent for the prebiotic mixture used in the bar, though the conflict of interest statement mentions those authors were not involved in subject recruitment, subject assessments, or statistical analysis of the data, showing they have tried to minimize the potential influence of this on study results.

Response: *Thank you for noting that study activities were carefully conducted to prevent any bias and that all potential conflicts have been clearly disclosed.*

(Comment 6) For the benefit of their patients, because of these promising results, I would strongly urge the authors to initiate a proper randomized controlled double blind trial to prove the efficacy of the product with high certainty. This should be feasible as authors already showed that only a relative low number of patients is needed to reach significance.

Response: *Thank you for your enthusiasm and for noting the importance of the results presented in this report. Indeed, applications to obtain the necessary funding to conduct trials have been submitted and placebo bars have been formulated to be used in randomized, controlled, double-blind trials in the future.*

(Comment 7) Given the importance of adherence, I would advise the authors to provide the evidence base for information (questionnaire?) on adherence, palatability etc. if available. It is not mentioned in the methods either.

Response: *We appreciate this suggestion. We have now included information about the questionnaire used to assess acceptability of the bar in the revised Manuscript with the full list of questions and outcomes (Table S1).*

(Comment 8) Abstract: As mentioned, one of the strengths of this study is the use of an in vitro model alongside clinical sampling, thus, it is worth mentioning the main results from the in vitro study in the abstract as well.

Response: *Thank you for this wonderful suggestion. The revised abstract explicitly states that this study includes both ex vivo fermentation and in vivo clinical results (line 71).*

(Comment 9) Abstract: A small grammatical error on page 4 lines 77-78: while relative abundance short-chain-fatty acids-producing bacteria were increased should be change to while the relative abundance of short-chain-fatty acid-producing bacteria was increased

Response: *Thank you for noting this oversight. The revised manuscript has been carefully proofed.*

(Comment 10) Introduction: The aims of the study should also list the in vitro study.

Response: *Thank you for this suggestion. The revised manuscript now explicitly states that this study includes both ex vivo fermentation and in vivo data (line 125).*

(Comment 11) Introduction: There is a repetition of "SCFA-producing" in the sentence listing the aim 2 on page 6 line 122

Response: *Thank you for noting this error. The revised manuscript has been carefully proofed.*

(Comment 12) Methods: In vitro study: There is a repetition of "slow fermentation" in the sentence on page 7 line 133-134. Could you clarify what it means in the context that they are both tolerable and allow to be delivered to the distal parts?

Response: *Thank you for noting this error. The revised manuscript has been carefully proofed. Regarding tolerability and delivery to the distal colon. Most commercially available fibers are rapidly fermented resulting in rapid accumulation of gas in the intestinal lumen leading to intestinal distention and discomfort. This characteristic makes high fiber consumption (i.e., doses needed to change the microbiota) poorly tolerated and unacceptable to individuals. However, knowledge of how chemical and physical structures influence fermentation were exploited in this study to develop a mixture of slow fermenting fibers. These slow-fermenting fibers produce gas slowly, over time and can be administered to humans potentially with few side effects than other fibers.^{31, 32} Ninety percent of the substrates used in the prebiotic mixture used in this study were found to be slow fermenting in our in vitro fermentation experiments (i.e., rice bran, potato starch, resistant maltodextrin). Indeed, participants largely reported the prebiotic intervention to be tolerable. The other important point is that rapidly fermented fibers are fermented by bacteria found in the small intestine and proximal portion of the large intestine; therefore, little fiber remains for bacteria in the*

colon. Slow fermenting fibers can be delivered to the distal large intestine ensuring that the benefits of SCFA production can be observed throughout the gastrointestinal tract. This important point is made in the revised manuscript (line 545).

(Comment 13) Methods: Could you clarify in how many replicates the experiments were conducted? Are there independent biological replicates or only technical replicates?

Response: Thank you for this noting important point. For the ex vivo stool fermentation, experimental (biological) triplicates were processed for SCFA analysis and two of these triplicates were analyzed independently for microbial community structure using 16S rRNA gene amplicon sequencing. Amplicons for each sample were sequenced on two sequential sequencing runs. For patient stool samples (no fermentation), single stool samples were processed for metagenome sequencing at baseline and after prebiotic intervention. No technical replicates were performed for these analyses. All bloodwork assays were run in duplicate. Details regarding the number of biological and technical duplicates for each assay are now included in the **Methods** section of the revised manuscript.

(Comment 14) Methods: Did you include any controls in the experiments? In Figure 1 there is a mention of a blank, but it is not described in the Methods.

Response: Thank you for this important question. For the clinical study, each subject served as their own control (i.e., a within subject comparison) which allowed us to understand how each individual changed in response to the prebiotic intervention. The within subject approach is commonly used in microbiota studies due to the highly variable nature of the microbiota between individuals. Each assay (e.g., ELISA) had internal controls (blanks) and for the ex vivo stool fermentation, a blank (tube with no added fiber) was included as a negative control. This information has been added to the revised **Methods**.

(Comment 15) Methods: Could you provide information for the bioinformatic processing of the 16S rRNA sequencing, including pipelines, decontamination, data cleaning procedures? Please also provide information on the statistical analyses of the relative abundances. Were these conducted compared to the control/blank group?

Response: Additional information about microbiota analysis is now included in the revised manuscript including methodology for data processing and analysis of the 16S rRNA sequencing. Relative abundance of bacteria from the ex vivo stool fermentation experiment were used only for visualization of clusters of promoted bacteria. Blanks (fermented fecal samples with no fiber added), were only used as a negative control for the ex vivo experiments and their sequencing data is not presented. Clustering for the heatmap was performed using Ward's algorithm based on Euclidean distances. This information has been added to the revised manuscript (line 337).

(Comment 16) Clinical study: Could you clarify if it was specified to the patients how and when they had to eat the prebiotic bar? I.e. what time of day, with food or not?

Response: Participants were asked to consume one bar on days 1-3 and two bars/day on days 4-10. Participants were instructed to consume the bar in the morning for the first three days, then in the morning and afternoon for the following week (they were not instructed to take it separately from meals) (line 215).

(Comment 17) Clinical study: Could you specify the faecal sample input amount to the DNA extraction protocol?

Response: Thank you for this suggestion. A total of 1ml of fermented stool sample was used for DNA extraction for 16S rRNA gene sequencing. A total of 250mg of stool was used per PD subject for DNA extractions to assess non-targeted shotgun metagenomics sequencing. This information is now included in the revised manuscript (line 144, line 151, line 239).

(Comment 18) Clinical study: In the paragraph on page 13 lines 273-276 could you specify if these analyses were also performed within-subjects, i.e. comparing data at baseline and post-prebiotic intervention?

Response: Thank you for noting this lack of clarity. The text in question has been updated to clarify that these evaluations were within subject (line 324).

(Comment 19) Clinical study: In regards of multiple comparison correction, the authors say on page 13 line 272 that "No adjustment for multiple testing was applied to this pilot study". Does this apply only to the demographic/clinical data or also for the analysis of the blood and SCFA markers?

Response: *This statement has been clarified in the revised manuscript. Adjustments for pairwise comparisons (q-values) were applied to SCFA data from ex vivo stool fermentation experiments and microbiome results (line 342, line 633).*

(Comment 20) Clinical study: Page 13 lines 284-286: there seem to be some words missing in the sentence. Do you mean that the NMDS plot was used to visualize the bacterial community structure of the samples?

Response: *Thank you for noting this mistake. The text in question has been revised (line 365).*

(Comment 21) Clinical study: Could you correct the sentence on page 13 line 287 to page 14 line 290? The Wilcoxon-signed rank test is a statistical hypothesis test and not a method for multiple comparison correction. What method did you use for adjusting for multiple comparisons here? Additionally, “matched samples” might be a better alternative to “paired analyses” in this sentence.

Response: *Thank you for noting this oversight. The text in the revised manuscript has been revised to state that the Wilcoxon-signed rank test was used to assess changes from baseline to after the prebiotic intervention (line 331) and states that a within subject analysis was conducted.*

(Comment 22) Clinical study: Analyses relating to differential relative abundances: please specify if you performed comparisons for all taxa identified in samples or only the top taxa presented in Fig 2c?

Response: *Thank you for this question. The revised manuscript now states that individual taxa and functional genes/pathways are shown for taxa with relative abundance greater than 0.1% (line 364). For visual representation purposes, only the mean relative abundance of taxa with greater than 1% average relative abundance are shown (noted in the legend of Figure 2).*

(Comment 23) Clinical study: What methods did you use to evaluate the outcome data distribution and homogeneity of variances?

Response: *Thank you for this question. We have now added the Permutational Analysis of Multivariate Dispersions (PERMDISP) to compare stool microbial community structures at baseline and after the prebiotic intervention. PERMDISP is a multivariate analogue of Levene’s test for homogeneity of multivariate variances that is commonly used in conjunction with PERMANOVA and assesses whether the dispersions between groups are significantly different. Results from the PERMDISP analysis indicate that PD group dispersions are similar to each other (i.e., non-significant). PERMDISP methods and results are included in the revised manuscript (Table S3). Additionally, the Shapiro-wilks test of normality was performed prior to performing pairwise comparisons (line 320).*

(Comment 24) Results, figures and tables: Could you explain the motivation for the correlation analyses starting from page 18 line 396? Although these results are interesting, currently it remains unclear what you were hoping to achieve by performing these analyses. In addition, could you explain if the correlation analyses were performed at baseline, post-prebiotic or taking all pre- and post-treatment samples into consideration?

Response: *Thank you for this important question. Correlation analysis was used to assess the relationship between microbiota/intestine specific outcomes and PD-relevant outcomes. This analysis is useful to identify specific taxa that may be of interest in future studies. Data used for Spearman correlation analysis was transformed using Log2 fold change from all PD participant’s baseline samples (n=20). This statement was added to the Methods (line 367).*

(Comment 25) Figure 1: page 24 lines 521-522: the authors say that “Bars denoted by a different letter indicate significant differences between treatment means ($p < 0.05$)”, however, it is not specified what each of the different letter mean. Please mention in the legend what the error bars represent on Fig 1b-e.

Response: *Thank you for this suggestion. The Figure 1 legend now states that error bars represent mean and standard deviation. To avoid confusion, the letters in Figure 1b-e were removed and the comprehensive results of pair-wise comparisons of total SCFA concentration after ex vivo stool fermentation are shown in Table S1.*

(Comment 26) Figure 2: Line 545: in “Non-metric MDS” explain the abbreviation in the legend. Differentially abundant bacteria before and after prebiotics: are these analyses adjusted for multiple comparisons? Given the number of comparisons performed, even as a proof-of-concept study, it would be crucial to correct for multiple testing. It appears from Table S1 that some adjustments for multiple testing was performed, but this should be made explicit in the Methods and main text figures. e-j: what was the basis of choosing to show

individual plots of these 6 taxa and leave out two other that had differential abundance (*Ruminococcus torques* and *bromii*)? It is a strength that individual datapoints are shown in d-k.

Response: Thank you for these questions. The abbreviation of “Non-metric MDS” has been addressed in Figure 2 legend. The text in question has been revised to now state, “Non-metric multidimensional scaling (NMDS) plots of the bacterial species community were used to visualize baseline and after prebiotic intervention data.” P-values were adjusted using the Benjamini-Hochberg method (q-values) for shotgun metagenomics microbiome analyses and these values are presented side-by-side in Tables S4, S5, S6. Of the species included in Figure 2e-j, these individual species are the most relevant to highlight based on previous studies in PD patients. The revised manuscript now includes species *Ruminococcus bromii* and *Ruminococcus torques* which have lower relative abundance after the prebiotic intervention to demonstrate that not all SCFA-producing bacteria are increased by the prebiotic intervention (**Figure 2k-l**).

(Comment 27) Table 2: I do not understand the values and IQR values provided for the tolerability data. It is a yes/no answer, correct, or a scale? As mentioned before, insights into the questions asked is relevant.

Response: Thank you for this important question. The questionnaire is a Likert scale between 0-10 for each question. We have modified the results to number and percent (n, %) for each response in the revised manuscript to improve clarity (**Table 2, Table S1**).

(Comment 28) Table S1: It might be better not to report p- and q-values mixed in the last column. As an alternative, the authors could report both the unadjusted p-values and FDR-corrected p-values (q-values) side by side in two columns as done for Table S2.

Response: Thank you for this suggestion. The Table has been revised to show a column for unadjusted p-values and the q-values to improve clarity (**Table S4, previously Table S1**). This same format was also used for **Tables S5 and S6**.

(Comment 29) Discussion: Page 22 line 486, sentence starting with “However, this is less likely because” the meaning of this sentence is unclear. There is also a double negative (“not non-fermenting”).

Response: The sentence in question has been revised for clarity as has the entire manuscript.

(Comment 30) Discussion: Page 22 line 490: what does it mean that samples were coded? Do you mean the laboratory work was performed blinded to the timepoint and patient group?

Response: Correct, sample analyses were conducted by lab personnel who were blinded to time point (baseline or after prebiotic intervention) and PD group (treated or de novo). This point has been clarified in the revised manuscript (**line 602**).

(Comment 31) Discussion: A major strength in the current study is the within-subjects design (i.e. always comparing baseline and post-prebiotic data of the same participant) and thus could be mentioned in the limitations/strength paragraph of the Discussion

Response: Thank you for noting the within subject design as a strength. This important point is now highlighted as a strength in the revised manuscript (**line 593**).

References

1. van Kessel SP, Bullock A, van Dijk G, et al. Parkinson's Disease Medication Alters Small Intestinal Motility and Microbiota Composition in Healthy Rats. *mSystems* 2022;7:e0119121.
2. Scheperjans F, Aho V, Pereira PA, et al. Gut microbiota are related to Parkinson's disease and clinical phenotype. *Mov Disord* 2015;30:350-8.
3. Pal GD, Shaikh M, Forsyth CB, et al. Abnormal lipopolysaccharide binding protein as marker of gastrointestinal inflammation in Parkinson disease. *Front Neurosci* 2015;9:306.
4. Chen SJ, Chi YC, Ho CH, et al. Plasma Lipopolysaccharide-Binding Protein Reflects Risk and Progression of Parkinson's Disease. *J Parkinsons Dis* 2021;11:1129-1139.
5. Hasegawa S, Goto S, Tsuji H, et al. Intestinal Dysbiosis and Lowered Serum Lipopolysaccharide-Binding Protein in Parkinson's Disease. *PLoS One* 2015;10:e0142164.
6. Forsyth CB, Shannon KM, Kordower JH, et al. Increased intestinal permeability correlates with sigmoid mucosa alpha-synuclein staining and endotoxin exposure markers in early Parkinson's disease. *PLoS One* 2011;6:e28032.

7. Schwartz A, Spiegel J, Dillmann U, et al. Fecal markers of intestinal inflammation and intestinal permeability are elevated in Parkinson's disease. *Parkinsonism Relat Disord* 2018;50:104-107.
8. Dumitrescu L, Marta D, Danau A, et al. Serum and Fecal Markers of Intestinal Inflammation and Intestinal Barrier Permeability Are Elevated in Parkinson's Disease. *Front Neurosci* 2021;15:689723.
9. Giron LB, Dweep H, Yin X, et al. Plasma Markers of Disrupted Gut Permeability in Severe COVID-19 Patients. *Front Immunol* 2021;12:686240.
10. Demir E, Ozkan H, Seckin KD, et al. Plasma Zonulin Levels as a Non-Invasive Biomarker of Intestinal Permeability in Women with Gestational Diabetes Mellitus. *Biomolecules* 2019;9.
11. Sturgeon C, Fasano A. Zonulin, a regulator of epithelial and endothelial barrier functions, and its involvement in chronic inflammatory diseases. *Tissue Barriers* 2016;4:e1251384.
12. Singh P, Silvester J, Chen X, et al. Serum zonulin is elevated in IBS and correlates with stool frequency in IBS-D. *United European Gastroenterol J* 2019;7:709-715.
13. Cummings JH, Pomare EW, Branch WJ, et al. Short chain fatty acids in human large intestine, portal, hepatic and venous blood. *Gut* 1987;28:1221-7.
14. Cummings JH, Macfarlane GT. The control and consequences of bacterial fermentation in the human colon. *J Appl Bacteriol* 1991;70:443-59.
15. Zhao L, Lou H, Peng Y, et al. Elevated levels of circulating short-chain fatty acids and bile acids in type 2 diabetes are linked to gut barrier disruption and disordered gut microbiota. *Diabetes Res Clin Pract* 2020;169:108418.
16. Yao L, Davidson EA, Shaikh MW, et al. Quantitative analysis of short-chain fatty acids in human plasma and serum by GC-MS. *Anal Bioanal Chem* 2022;414:4391-4399.
17. Ryu JK, Kim SJ, Rah SH, et al. Reconstruction of LPS Transfer Cascade Reveals Structural Determinants within LBP, CD14, and TLR4-MD2 for Efficient LPS Recognition and Transfer. *Immunity* 2017;46:38-50.
18. Schumann RR, Kirschning CJ, Unbehauen A, et al. The lipopolysaccharide-binding protein is a secretory class 1 acute-phase protein whose gene is transcriptionally activated by APRF/STAT/3 and other cytokine-inducible nuclear proteins. *Mol Cell Biol* 1996;16:3490-503.
19. Zhou Z, Xu MJ, Gao B. Hepatocytes: a key cell type for innate immunity. *Cell Mol Immunol* 2016;13:301-15.
20. Chen KF, Chaou CH, Jiang JY, et al. Diagnostic Accuracy of Lipopolysaccharide-Binding Protein as Biomarker for Sepsis in Adult Patients: A Systematic Review and Meta-Analysis. *PLoS One* 2016;11:e0153188.
21. Ghanim H, Abuaysheh S, Sia CL, et al. Increase in plasma endotoxin concentrations and the expression of Toll-like receptors and suppressor of cytokine signaling-3 in mononuclear cells after a high-fat, high-carbohydrate meal: implications for insulin resistance. *Diabetes Care* 2009;32:2281-7.
22. Liu X, Lu L, Yao P, et al. Lipopolysaccharide binding protein, obesity status and incidence of metabolic syndrome: a prospective study among middle-aged and older Chinese. *Diabetologia* 2014;57:1834-41.
23. Cani PD, Osto M, Geurts L, et al. Involvement of gut microbiota in the development of low-grade inflammation and type 2 diabetes associated with obesity. *Gut Microbes* 2012;3:279-88.
24. Cani PD, Possemiers S, Van de Wiele T, et al. Changes in gut microbiota control inflammation in obese mice through a mechanism involving GLP-2-driven improvement of gut permeability. *Gut* 2009;58:1091-103.
25. Theede K, Holck S, Ibsen P, et al. Level of Fecal Calprotectin Correlates With Endoscopic and Histologic Inflammation and Identifies Patients With Mucosal Healing in Ulcerative Colitis. *Clin Gastroenterol Hepatol* 2015;13:1929-36 e1.
26. Ricciuto A, Griffiths AM. Clinical value of fecal calprotectin. *Crit Rev Clin Lab Sci* 2019;56:307-320.
27. Mulak A, Koszewicz M, Panek-Jeziorna M, et al. Fecal Calprotectin as a Marker of the Gut Immune System Activation Is Elevated in Parkinson's Disease. *Front Neurosci* 2019;13:992.
28. Schroder H, Fito M, Estruch R, et al. A short screener is valid for assessing Mediterranean diet adherence among older Spanish men and women. *J Nutr* 2011;141:1140-5.

29. David LA, Maurice CF, Carmody RN, et al. Diet rapidly and reproducibly alters the human gut microbiome. *Nature* 2014;505:559-63.
30. van Kessel SP, Frye AK, El-Gendy AO, et al. Gut bacterial tyrosine decarboxylases restrict levels of levodopa in the treatment of Parkinson's disease. *Nat Commun* 2019;10:310.
31. Rumpagaporn P, Reuhs BL, Kaur A, et al. Structural features of soluble cereal arabinoxylan fibers associated with a slow rate of in vitro fermentation by human fecal microbiota. *Carbohydr Polym* 2015;130:191-7.
32. Kaur A, Rose DJ, Rumpagaporn P, et al. In vitro batch fecal fermentation comparison of gas and short-chain fatty acid production using "slowly fermentable" dietary fibers. *J Food Sci* 2011;76:H137-42.

Reviewers' Comments:

Reviewer #1:

Remarks to the Author:

This reviewer appreciates the thoughtful responses to this and the other reviewer comments. This is an important study, whose promising results (while indeed still in the proof-of-concept phase) will be very beneficial for the field.

Reviewer #2:

Remarks to the Author:

The authors have addressed all important points to my satisfaction. The interpretation of the biomarkers is now much more nuanced and 'putative' is used consistently throughout the manuscript in case of the marker zonulin-1.

There is one final comment which relates to the new ex vivo experiment using stool from PD patients and controls.

The new ex vivo fermentation experiments are very insightful and relevant for GPs and patients. They should be included as a figure in the manuscript and not as a supplement for the reasons outlined below. The individual SCFA should be provided to that it is clear whether a specific or all SCFAs are numerically lower in the naïve state (without substrate).

Reasons to include the data as a figure: First, these data demonstrate that PD patients can in principle produce SCFA in comparable quantities as healthy control subjects indicating that their microbiota is not impaired in terms of functionality if substrate is available. This point should in my view also be included in the Discussion.

Second, the relative difference between healthy subjects and PD patients in the naïve state (without substrate) is rather subtle although being significant. Much more important seems to be the potential of the PD microbiota to produce high quantities of SCFA once substrate has been made available. This is a rather hopeful finding which deserves more attention (a figure in the MS).

It is also an extremely important finding for both patients and medical doctors. The problem is thus not the microbiota per se but the availability of substrate.

Reviewer #3:

Remarks to the Author:

Great findings

I would stay away from a signature microbiome on Parkinson's as data coming that may contradict this especially as we are all different and therefore it's hard to figure out a microbiome signature. On the whole good article. Gives hope to patients but is far from being an answer to this very complex disease.

Reviewer #4:

Remarks to the Author:

We were pleased to see all our comments addressed in the new manuscript.

We only observed one minor error;

Line 362 "Differences in the relative abundance of individual taxa and functional genes/pathways between PD groups were assessed for significance with relative abundance greater than 0.1%"
It seems you didn't assess differences between PD groups but rather between baseline and after probiotic, this should be corrected (by removing "between PD groups").

We have no further comments.

Reviewer #5:

Remarks to the Author:

I have been asked to provide a statistical review of this report of a proof of concept study into a prebiotic intervention for Parkinson's disease.

While the study is in general well reported, and I look forward the report of the subsequent randomised trial, I have several concerns that should be addressed before the report is suitable for publication.

- 1) The authors report findings of an experiment whereby stool samples from 10 healthy controls and 10 PD patients are fermented with different fibres, is the recruitment of these participants described in the methods?
- 2) In figure 1a there are four columns under each fibre, but the methods refer to three biological and three technical replicates. What do the columns correspond to?
- 3) Also in figure 1a, what does the scale bar (numbers from -2 to +2 represent)?
- 4) In figure 1b there is no statistical analysis to support the claims made for the effect of different fibres on SCFA production. The authors could show individual data points (which in general is done very well in the rest of the report) being mindful not to conflate technical with biological variation.
- 5) Line 415 - there is not enough evidence that the intervention does not cause side effects, you can only say that no substantial side effects were seen in this group.
- 6) The test for a change in overall gastro symptom scores is not statistically significant. A regression analysis testing for a change in symptom scores between groups is only likely to pick up the fact that the existing PD group had higher scores initially. However as a proof of concept study the authors might mention a possible improvement in symptoms ($p=0.09$), but be cautious about the interpretation and of the subgroup analysis.
- 7) Throughout, where a p-value is mentioned (in the text or a figure caption) the test used to generate it should be mentioned.
- 8) Although the study was not designed to test UPDRS scores this finding is given the most prominence in the discussion and the abstract, where it is claimed that the "10 days of prebiotic intervention improved the UPDRS score". Given the uncontrolled nature of the study this causal claim is not supported.
- 9) Given the prominence and potential importance of the UPDRS finding the individual data points for each participant should be given (and ideally published as a data file), as is the case with most of the other findings in the paper.
- 10) Throughout, the authors interpret p values of less than 0.05 as being statistically significant, even though the q-values are very high. Given the very high number of tests conducted many of these

$p < 0.05$ findings are likely to be positives, and authors should restrict themselves to discussing calling results at a q -value threshold.

11) The use of Wilcoxon paired tests is acceptable, and the presentation of individual changes in bacterial taxon prevalence is good. There is no consensus in the literature as to how differential abundance in microbiome data should be analysed, and this presentation does allow for an interpretation of which taxa are likely increased or decreased. But again, it would help for the authors to publish the taxon count table so that others could run their own analyses if needed.

12) I couldn't find any result in the paper corresponding to the claim that the intervention "improved gastrointestinal symptoms including constipation in those who had constipation prior to the start of the study". In any case such a finding in an uncontrolled study would be likely attributable to 'regression to the mean' whereby a subset of a group selected on the basis of having a symptom at baseline resolve that symptom, giving the appearance of a treatment effect. The full data and analysis supporting this claim should be added or the claim removed.

13) Multiple testing corrections should be applied to p -values in table 4 as presumably a lot of different hypotheses were tested.

14) The PERMANOVA test should have permutations restricted to respect the pairing (pre-post) intervention of the samples. It would be helpful for the authors to join the points corresponding to individual participants.

15) Although statistically significant, the authors should point out that the actual difference in NFI is extremely small.

Point by point responses are detailed below and changes in the manuscript (i.e., additions/clarifications that address reviewer comments) are indicated in **blue**.

Reviewer 1

(Comment 1) This reviewer appreciates the thoughtful responses to this and the other reviewer comments. This is an important study, whose promising results (while indeed still in the proof-of-concept phase) will be very beneficial for the field.

***Response:** We appreciate the kind words and look forward to sharing these results with the research community.*

Reviewer #2

(Comment 1) The authors have addressed all important points to my satisfaction. The interpretation of the biomarkers is now much more nuanced and 'putative' is used consistently throughout the manuscript in case of the marker zonulin-1.

***Response:** Thank you for the kind words and we appreciate the constructive feedback during the prior review.*

(Comment 2) There is one final comment which relates to the new ex vivo experiment using stool from PD patients and controls. The new ex vivo fermentation experiments are very insightful and relevant for GPs and patients. They should be included as a figure in the manuscript and not as a supplement for the reasons outlined below. The individual SCFA should be provided to that it is clear whether a specific or all SCFAs are numerically lower in the native state (without substrate). Reasons to include the data as a figure: First, these data demonstrate that PD patients can in principle produce SCFA in comparable quantities as healthy control subjects indicating that their microbiota is not impaired in terms of functionality if substrate is available. This point should in my view also be included in the Discussion. Second, the relative difference between healthy subjects and PD patients in the native state (without substrate) is rather subtle although being significant. Much more important seems to be the potential of the PD microbiota to produce high quantities of SCFA once substrate has been made available. This is a rather hopeful finding which deserves more attention (a figure in the MS). It is also an extremely important finding for both patients and medical doctors. The problem is thus not the microbiota per se but the availability of substrate.

***Response:** We thank the reviewer for this suggestion. Figure S1 has been moved to the main manuscript, now Figure 2, and subsequent figure numbers in the manuscript have been updated to reflect this addition.*

Reviewer #3

(Comment 1) Great findings

***Response:** We appreciate the kind words and look forward to sharing these results with the research community.*

(Comment 2) I would stay away from a signature microbiome on Parkinson's as data coming that may contradict this especially as we are all different and therefore it's hard to figure out a microbiome signature.

***Response:** Indeed, we agree and have acknowledged that there is no specific microbiota signature for PD but have noted some of the most commonly reported changes in the microbiota (**line 99-107**).*

(Comment 3) On the whole good article. Gives hope to patients but is far from being an answer to this very complex disease.

***Response:** Thank you for this positive comment. We too agree that PD is highly complex which underscores the need for additional research.*

Reviewer #4

(Comment 1) We were pleased to see all our comments addressed in the new manuscript.

***Response:** We appreciate the kind words and are appreciative of the constructive feedback in the prior review.*

(Comment 2) We only observed one minor error; Line 362 "Differences in the relative abundance of individual taxa and functional genes/pathways between PD groups were assessed for significance with relative

abundance greater than 0.1%??? It seems you didn't assess differences between PD groups but rather between baseline and after probiotic, this should be corrected (by removing "between PD groups???").

Response: Thank you for noting this oversight. The statement of "between PD groups" was removed and replaced with "between baseline and after probiotic intervention" (line 375).

(Comment 3) We have no further comments.

Response: We appreciate the constructive feedback in the prior review that helped improve the manuscript.

Reviewer #5

(General Comment) I have been asked to provide a statistical review of this report of a proof of concept study into a probiotic intervention for Parkinson's disease. While the study is in general well reported, and I look forward the report of the subsequent randomised trial, I have several concerns that should be addressed before the report is suitable for publication.

Response: We appreciate the kind words and look forward to sharing these results with the research community.

(Comment 1) The authors report findings of an experiment whereby stool samples from 10 healthy controls and 10 PD patients are fermented with different fibres, is the recruitment of these participants described in the methods?

Response: Thank you for noting this important point. Indeed, the ex vivo analysis preceded the in vivo study, thus the stool samples were from different participants than those included in the in vivo study. The revised manuscript now includes a more thorough description of the participant stool used. "**Participants:** Prebiotic fibers (inulin, resistant starch, resistant maltodextrin, and rice bran) were incubated with human stool from healthy control (n=10) or PD (n=10) participants. Stool samples used for this ex vivo analysis were obtained from an IRB-approved repository. All participants donating samples to the repository signed the Rush University Medical Center (RUMC) Institutional Review Board approved informed consent form (ORA#: 07100403) and were recruited from the Chicagoland area. The healthy control and PD participants used for the ex vivo study were not enrolled into the in vivo probiotic study (described below)." (lines 136-142)

(Comment 2) In figure 1a there are four columns under each fibre, but the methods refer to three biological and three technical replicates. What do the columns correspond to? Also in figure 1a, what does the scale bar (numbers from -2 to +2 represent)?

Response: Thank you for these essential questions. Experimental triplicates were run, two of which were used for sequencing which was performed twice (totaling four, each represented in a separate column in Figure 1). The scale bar represents Z-scores of relative abundance normalized by row. These important pieces of information have been added to the figure legend and figure. (Figure 1, lines 666-668).

(Comment 3) In figure 1b there is no statistical analysis to support the claims made for the effect of different fibres on SCFA production. The authors could show individual data points (which in general is done very well in the rest of the report) being mindful not to conflate technical with biological variation.

Response: Thank you for noting this oversight. The individual points have now been included on each graph (Figure 1), the overall ANOVA value is provided in the text (lines 397-398), and significant between group comparisons are indicated on the graph which are defined in the figure legend (lines 671-674). Group averages and specific between group differences (post hoc Tukey) can be found in Table S1.

(Comment 4) Line 415 - there is not enough evidence that the intervention does not cause side effects, you can only say that no substantial side effects were seen in this group.

Response: We agree and have changed the text accordingly, "the probiotic fiber mixture did not cause substantial or clinically pertinent gastrointestinal side effects." (line 434).

(Comment 5) The test for a change in overall gastro symptom scores is not statistically significant. A regression analysis testing for a change in symptom scores between groups is only likely to pick up the fact that the existing PD group had higher scores initially. However as a proof of concept study the authors might mention a possible improvement in symptoms ($p=0.09$), but be cautious about the interpretation and of the subgroup analysis.

Response: Thank you for your attention to these details. We too are cautious about the interpretation and as such have framed the discussion with the preface that this is a proof-of-concept study and avoided any allusion to causality but simply state the observation that GI symptoms and UPDRS are impacted. Moreover, we address that the non-blinded design could have impacted outcomes (*lines 626-627*) and we emphasize that future placebo-controlled studies are needed (*lines 622-625, 626-627, 634-635*).

(Comment 6) Throughout, where a p-value is mentioned (in the text or a figure caption) the test used to generate it should be mentioned.

Response: We thank the reviewer for this suggestion. Throughout the manuscript (text and figure legends), we have now added the statistical test used to generate the p-value for each comparison.

(Comment 7) Although the study was not designed to test UPDRS scores this finding is given the most prominence in the discussion and the abstract, where it is claimed that the "10 days of prebiotic intervention improved the UPDRS score". Given the uncontrolled nature of the study this causal claim is not supported.

Response: Thank you for noting this important point. Accordingly, we have modified both the abstract and the discussion to retain the clinical observation but remove any insinuation of causality from microbiome changes: "This proof-of-concept study demonstrated that a prebiotic intervention is tolerated by PD patients and can alter the microbiome with a signal of possible impact on outcomes relevant to PD." (*line 85-87*)

(Comment 8) Given the prominence and potential importance of the UPDRS finding the individual data points for each participant should be given (and ideally published as a data file), as is the case with most of the other findings in the paper.

Response: We appreciate this request and have included a Supplementary Data File to allow readers to view individual data points. This Supplementary Data File is now referenced in the Results section.

(Comment 9) Throughout, the authors interpret p values of less than 0.05 as being statistically significant, even though the q-values are very high. Given the very high number of tests conducted many of these $p < 0.05$ findings are likely to be positives, and authors should restrict themselves to discussing calling results at a q-value threshold.

Response: We appreciate this attention to detail. Given the small sample size we believe that a p -value < 0.05 (even in the absence of a significant q -value) is worth noting. However, we agree that there is a possibility of false positives. To address this important point, we have omitted the use of the word "significant" and instead refer readers to the data and analyses so the reader can derive their own interpretation.

(Comment 10) The use of Wilcoxon paired tests is acceptable, and the presentation of individual changes in bacterial taxon prevalence is good. There is no consensus in the literature as to how differential abundance in microbiome data should be analyzed, and this presentation does allow for an interpretation of which taxa are likely increased or decreased. But again, it would help for the authors to publish the taxon count table so that others could run their own analyses if needed.

Response: Thank you for noting that the use of the Wilcoxon paired test is an acceptable approach and we agree that the taxon counts may be of value to readers. Accordingly, we have added the taxonomic and functional gene pathway biological observation matrices (BIOMS) in the Supplementary Data File. Additionally, raw sequence data were deposited into the appropriate database for readers to obtain if they wish as stated in the manuscript: "Raw sequence data (FASTQ files) were deposited in the National Center for Biotechnology Information (NCBI) Sequence Read Archive (SRA), under the BioProject identifier PRJNA756556."

(Comment 11) I couldn't find any result in the paper corresponding to the claim that the intervention improved gastrointestinal symptoms including constipation in those who had constipation prior to the start of the study. In any case such a finding in an uncontrolled study would be likely attributable to regression to the mean whereby a subset of a group selected on the basis of having a symptom at baseline resolve that symptom, giving the appearance of a treatment effect. The full data and analysis supporting this claim should be added or the claim removed.

Response: Indeed, the statement was based on the concept that constipation can only be improved in those with constipation (floor effect). However, the statement in question has been deleted to best align with the data presented in the manuscript.

(Comment 12) Multiple testing corrections should be applied to p-values in table 4 as presumably a lot of different hypotheses were tested.

Response: *Thank you for this suggestion. A correction for multiple testing (i.e., q-values) was conducted (using the Benjamini-Hochberg method) and were added to **Table 4**, and a descriptive statement reflecting the analysis is now included in the Methods (**lines 378**).*

(Comment 13) The PERMANOVA test should have permutations restricted to respect the pairing (pre-post) intervention of the samples. It would be helpful for the authors to join the points corresponding to individual participants.

Response: *We agree that this visualization would be helpful to readers. Accordingly, we have revised **Figure 3b** to include the participant ID to depict the pre-post prebiotic intervention pairing for the total microbial community structure distances for each PD subject (**Figure 3b**).*

(Comment 14) Although statistically significant, the authors should point out that the actual difference in NFI is extremely small.

Response: *Indeed, thank you for this suggestion. There is a significant reduction, but as you note we do not fully understand the biological relevance of this finding. We now note that this a subtle change (**line 545**).*

Reviewers' Comments:

Reviewer #5:

Remarks to the Author:

Thank you for your revision. The presentation of the statistical analysis and methods is now much improved, and the additional of the individual data file is enormously helpful in interpreting the results.

However I have two outstanding comments. First, the interpretation of the findings in the abstract and discussion is still overstated given that this is a very small uncontrolled open label pre-post study. For example, the statement in the abstract

"Ten days of prebiotic fiber intervention improved gastrointestinal and PD motor symptoms"

is much too strong, given that the change in gastrointestinal symptoms was not statistically significant, and there is only a correlational association between the administration of the intervention with the change in motor symptoms and we cannot attribute this to the treatment. This should be addressed throughout the manuscript.

Second, it is certain that many of the specific bacterial called as significantly altered by the treatment will be false positive findings. It is reasonable to say there is some evidence that the microbiome is affected by the treatment given the shift in proteobacteria, but statements like:

"The relative abundance of putative SCFA-producing species was increased by the prebiotic intervention including the SCFA-producing species *Faecalibacterium prausnitzii* (RA:464 baseline 0.00-19.50%, after intervention 0.00-24.57%)"

clearly suggest that the intervention is responsible for an increase in this bacteria even though the q-value for this is $q=0.313$!

The authors can point out in these cases that some bacteria were higher in abundance after the intervention than before, but should explicitly mention that these changes are not statistically significant and cannot be attributed to the intervention.

Point by point responses are detailed below and changes in the manuscript (i.e., additions/clarifications that address reviewer comments) are indicated in **blue**.

Reviewer #5 Comments

(Comment 1) Thank you for your revision. The presentation of the statistical analysis and methods is now much improved, and the additional of the individual data file is enormously helpful in interpreting the results. However I have two outstanding comments.

Response: Thank you for the suggestions and we agree these changes will be beneficial for the readers.

(Comment 2) First, the interpretation of the findings in the abstract and discussion is still overstated given that this is a very small uncontrolled open label pre-post study. For example, the statement in the abstract "Ten days of prebiotic fiber intervention improved gastrointestinal and PD motor symptoms" is much too strong, given that the change in gastrointestinal symptoms was not statistically significant, and there is only a correlational association between the administration of the intervention with the change in motor symptoms and we cannot attribute this to the treatment. This should be addressed throughout the manuscript.

Response: We agree that this is a small, uncontrolled, open label study. However, there was a statistically significant difference between Total Gastrointestinal Score for Treated PD Participants before and after prebiotic treatment (Table 2, $p=0.01$). But we agree that that this is only significant in treated PD participants and could be due to a placebo effect. As such, we have revised the abstract (and the manuscript) to not overstate or overinterpret the outcomes.

(Comment 3) Second, it is certain that many of the specific bacterial called as significantly altered by the treatment will be false positive findings. It is reasonable to say there is some evidence that the microbiome is affected by the treatment given the shift in proteobacteria, but statements like: "The relative abundance of putative SCFA-producing species was increased by the prebiotic intervention including the SCFA-producing species *Faecalibacterium prausnitzii* (RA:464 baseline 0.00-19.50%, after intervention 0.00-24.57%)" clearly suggest that the intervention is responsible for an increase in this bacteria even though the q-value for this is $q=0.313$! The authors can point out in these cases that some bacteria were higher in abundance after the intervention than before, but should explicitly mention that these changes are not statistically significant and cannot be attributed to the intervention.

Response: We understand the hesitation to refer to changes in specific taxa as significant without a q-value < 0.05 . We have refrained from using the word significant when discussing these changes and have clearly indicated that these do not meet the stringent level of criteria for q significance. Additionally, both the p and q values are now included on the figures when available.